# Transcriptional control of subtype switching ensures adaptation and growth of pancreatic cancer

Christina R Adams[1], Htet Htwe Htwe[1], Timothy Marsh[2], Aprilgate L Wang[1], Megan L Montoya[1], Lakshmipriya Subbaraj[3], Aaron D Tward[3,4], Nabeel Bardeesy[5], Rushika M Perera[1,2,4]*

[1]Department of Anatomy, University of California, San Francisco, San Francisco, United States; [2]Department of Pathology, University of California, San Francisco, San Francisco, United States; [3]Department of Otolaryngology, University of California, San Francisco, San Francisco, United States; [4]Helen Diller Family Comprehensive Cancer Center, University of California, San Francisco, San Francisco, United States; [5]Massachusetts General Hospital Cancer Center, Harvard Medical School, Boston, United States

**Abstract** Pancreatic ductal adenocarcinoma (PDA) is a heterogeneous disease comprised of a basal-like subtype with mesenchymal gene signatures, undifferentiated histopathology and worse prognosis compared to the classical subtype. Despite their prognostic and therapeutic value, the key drivers that establish and control subtype identity remain unknown. Here, we demonstrate that PDA subtypes are not permanently encoded, and identify the GLI2 transcription factor as a master regulator of subtype inter-conversion. GLI2 is elevated in basal-like PDA lines and patient specimens, and forced GLI2 activation is sufficient to convert classical PDA cells to basal-like. Mechanistically, GLI2 upregulates expression of the pro-tumorigenic secreted protein, Osteopontin (OPN), which is especially critical for metastatic growth in vivo and adaptation to oncogenic KRAS ablation. Accordingly, elevated GLI2 and OPN levels predict shortened overall survival of PDA patients. Thus, the GLI2-OPN circuit is a driver of PDA cell plasticity that establishes and maintains an aggressive variant of this disease.
DOI: https://doi.org/10.7554/eLife.45313.001

*For correspondence:
rushika.perera@ucsf.edu

**Competing interests:** The authors declare that no competing interests exist.

## Introduction

Pancreatic ductal adenocarcinoma (PDA) is projected to become the second leading cause of cancer-related deaths by 2030 (*Rahib et al., 2014*), and has a 5-year survival rate of <10%. Genomic and transcriptomic analyses have revealed important insights into the underlying biology of this disease. The most common genetic alterations include activating mutations in *KRAS* in ~95% of PDA and inactivating mutations or deletions of *TP53*, *CDKN2A*, and *SMAD4* in 50–70% (*Jones et al., 2008*; *Biankin et al., 2012*; *Ryan et al., 2014*; *Waddell et al., 2015*; *Witkiewicz et al., 2015*). Recently, transcriptional profiling from resected PDA specimens has identified two main subtypes with distinct molecular features, termed classical and basal-like (*Collisson et al., 2011*; *Moffitt et al., 2015*; *Bailey et al., 2016*). Classical PDA is enriched for expression of epithelial differentiation genes, whereas basal-like PDA is characterized by laminin and basal keratin gene expression, stem cell and epithelial-to-mesenchymal transition (EMT) markers, analogous to the basal subtypes previously defined in bladder and breast cancers (*Perou et al., 2000*; *Parker et al., 2009*; *Curtis et al., 2012*; *Cancer Genome Atlas Research Network, 2014*; *Damrauer et al., 2014*). Importantly, basal-like subtype tumors display poorly differentiated histological features and

correlate with markedly worse prognosis (*Moffitt et al., 2015*; *Cancer Genome Atlas Research Network, 2017*; *Aung et al., 2018*). These subtypes are preserved in different experimental models of PDA including organoids (*Boj et al., 2015*; *Huang et al., 2015*; *Seino et al., 2018*), cell line cultures (*Collisson et al., 2011*; *Moffitt et al., 2015*; *Martinelli et al., 2017*), and a genetically engineered mouse (GEM) model of PDA in which ablation of oncogenic Kras resulted in subtype conversion (*Kapoor et al., 2014*). However, the identity of key factors responsible for establishing and maintaining subtype specificity and how these programs integrate with pathways known to be deregulated in PDA remain largely unknown.

The Hedgehog (Hh) pathway is activated in PDA and has been found to play important and complex roles in PDA pathogenesis (*Morris et al., 2010*). Whereas the developing and normal adult pancreas lack expression of Hh pathway ligands, the Sonic Hedgehog (SHH) and Indian Hedgehog (IHH) ligands are prominently induced in the pancreatic epithelium upon injury and throughout PDA development, from early precursor pancreatic intraepithelial neoplasia (PanIN) to invasive disease (*Berman et al., 2003*; *Thayer et al., 2003*; *Prasad et al., 2005*; *Nolan-Stevaux et al., 2009*). The neoplastic cells and stromal fibroblasts also express the Hh receptor Smoothened (SMO) and the Glioma-associated oncogene homology (GLI) transcription factors – GLI1 and GLI2, which mediate Hh signaling downstream of SMO, and GLI3 which functions as a transcriptional repressor (*Hui and Angers, 2011*; *Robbins et al., 2012*). While *Smo* deletion in the pancreatic epithelium has no effect on mutant KRAS-driven PDA in GEM models, studies from our group and others reveal a surprising role for SHH in restraining cancer growth (*Lee et al., 2014*; *Mathew et al., 2014*; *Rhim et al., 2014*; *Liu et al., 2016*). By contrast, several lines of evidence indicate that activation of GLI transcription factors in the pancreatic epithelium is required for oncogenesis in PDA (*Dennler et al., 2007*; *Ji et al., 2007*; *Nolan-Stevaux et al., 2009*; *Rajurkar et al., 2012*; *Xu et al., 2012*). First, pancreas-specific transgenic over-expression of the *Gli3* repressor attenuates PDA progression (*Rajurkar et al., 2012*). Second, forced *Gli2* over-expression cooperates with oncogenic Kras to promote aggressive poorly differentiated tumors (*Pasca di Magliano et al., 2006*). While much emphasis has been placed on ligand-mediated activation of Hh signaling in stromal cells within PDA, the tumor cell-autonomous roles of GLI proteins have remained unclear. In other cancer settings, GLI proteins have been implicated in cell cycle progression, activation of pro-survival programs and a cancer-associated EMT (*Yoon et al., 2002*; *Alexaki et al., 2010*; *Das et al., 2013*; *Han et al., 2015*; *Neelakantan et al., 2017*). We therefore explored a role for GLI transcription factors in regulating PDA subtype identity and inter-conversion. We show that GLI2 functions as a master regulator of the basal-like subset of PDA, and define the secreted pro-inflammatory cytokine, Osteopontin (OPN), as a critical downstream target mediating this program. Moreover, our findings indicate that a GLI-OPN axis can functionally substitute for proliferative and survival cues regularly provided by oncogenic Kras in PDA. Together, these data identify GLI2 as an Hh-independent, cell autonomous driver of an aggressive variant of PDA and illuminate the complex role of Hh pathway components in PDA tumorigenesis.

## Results

### Expression of Hh ligands and GLI transcription factors are anti-correlated and predict survival outcomes in PDA

To explore the relationship between the Hh pathway components in PDA, we first determined the expression levels of SHH and GLI family transcription factors (GLI1, GLI2, GLI3) in a panel of 14 human PDA cell lines using validated antibodies (*Figure 1—figure supplement 1A,B*). We found that all PDA cell lines expressed Hh pathway proteins to varying degrees. GLI1 and GLI3 expression were restricted to two- and one-cell line respectively, while GLI2 was readily detectable in 10/14 lines. Moreover, high levels of SHH were observed in 7/14 cell lines, while the remainder showed low or undetectable levels of expression (*Figure 1A*). No significant differences in the level of the SMO receptor were observed across the panel. Notably, cell lines expressing the highest levels of GLI2 (KP4, MiaPaca and Panc0327 cells) displayed the lowest levels of SHH expression (*Figure 1A*; asterisk). Conversely, GLI expression was uniformly low in lines with high levels of SHH. Accordingly, GLI responsive luciferase reporter assay (*Sasaki et al., 1997*) demonstrated that GLI transcriptional activity was inversely correlated with SHH expression and positively correlated with GLI expression

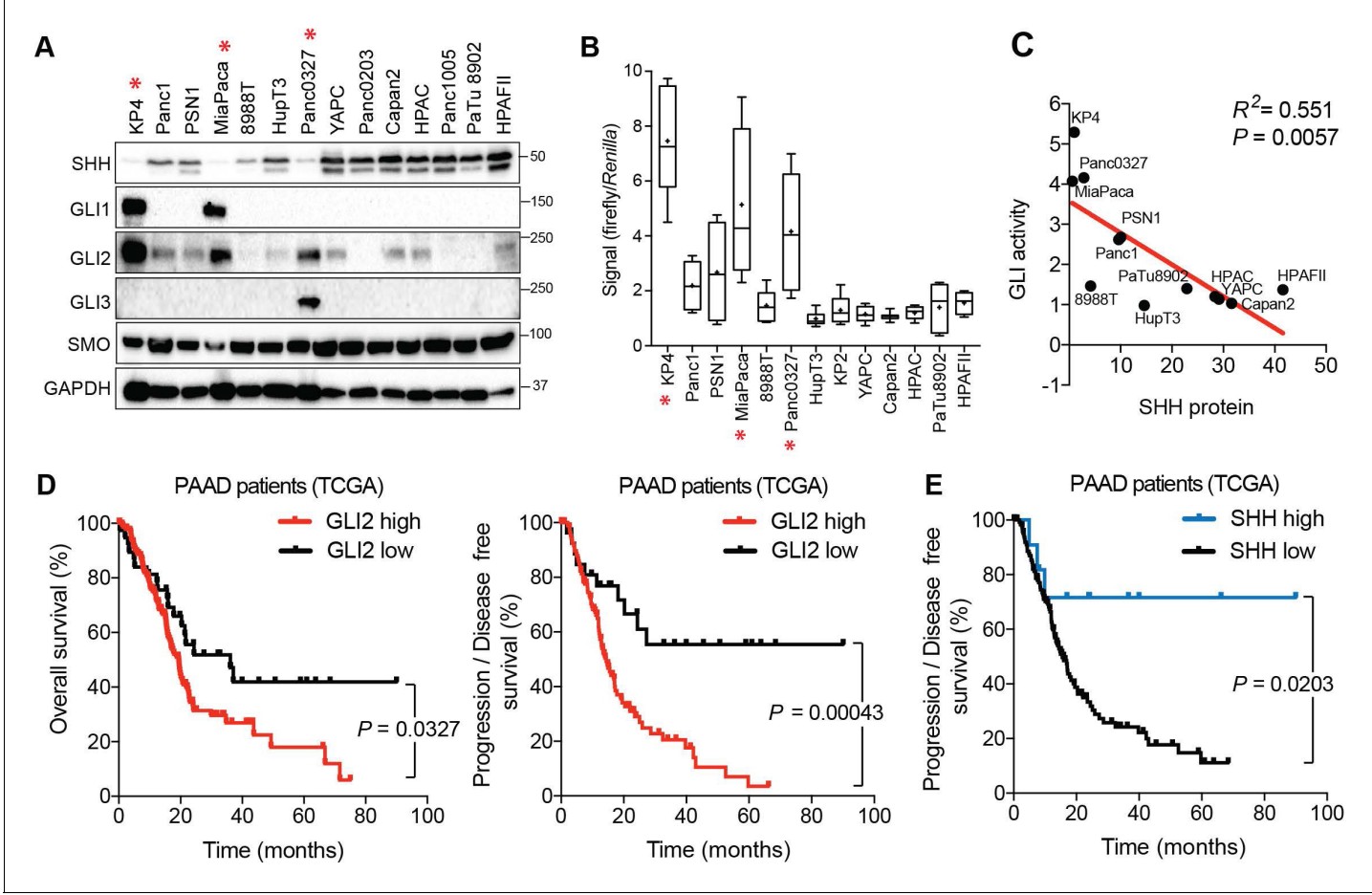

**Figure 1.** Expression and activity of GLI proteins is anti-correlated with Hh ligand levels in PDA. (**A**) Immunoblots showing expression of the indicated proteins in 14 human PDA cell lines. Cell lines denoted with an asterisk represent GLI^hi/Hh^lo lines. (**B**) The indicated cell lines were transfected with the 8 × 3'Gli-binding site luciferase plasmid and luciferase activity was measured 48 hr post-transfection. For each box-and-whisker plot, center asterisk indicates the mean of n = 3 experiments with a total of 10 independently transfected cultures for each cell line. (**C**) GLI activity as measured by GLI luciferase assay (**B**) is anti-correlated with SHH protein levels (**A**). Linear regression line is shown in red along with corresponding statistics. (**D,E**) High expression of *GLI2* predicts shorter overall (D: left; GLI2 high n = 147, GLI2 low n = 38) and disease-free (D: right; GLI2 high n = 114, GLI2 low n = 27) survival while high *SHH* expression predicts extended disease-free survival (E: SHH high n = 11, SHH low n = 130) for patients with PDA from The Cancer Genome Atlas (TCGA). Data from 185 patients. p-Value calculated by Log-rank test.

DOI: https://doi.org/10.7554/eLife.45313.002

The following source data and figure supplements are available for figure 1:

**Source data 1.** GLI proteins are expressed in basal-like PDA.

DOI: https://doi.org/10.7554/eLife.45313.005

**Figure supplement 1.** GLI proteins are expressed in basal-like PDA.

DOI: https://doi.org/10.7554/eLife.45313.003

**Figure supplement 2.** Cox proportional hazards regression models for predictors of overall survival and progression-free survival.

DOI: https://doi.org/10.7554/eLife.45313.004

(*Figure 1B,C*), such that all SHH^hi cell lines lacked basal reporter activity (*Figure 1B*), whereas the GLI^hi KP4, MiaPaca and Panc0327 lines exhibited the highest levels of activity (*Figure 1B,C*). Luciferase activity was specific to GLI proteins as shRNA-mediated knockdown of GLI2 or treating cells with GANT61 inhibitor, which blocks GLI binding to DNA (*Lauth et al., 2007*), suppressed GLI-driven luciferase activity (*Figure 1—figure supplement 1C,D*).

The anti-correlation between GLI2 and SHH was further corroborated by analyzing RNA-sequencing (RNA-seq) data of resected PDA specimens from The Cancer Genome Atlas (TCGA) study (*Cancer Genome Atlas Research Network, 2017*). Samples were binned into two groups based on

whether *GLI2* expression levels were higher or lower than the mean mRNA expression within the sample set (see Materials and methods for details). Consistent with the cell line data, tumors expressing high levels of *GLI2* mRNA had significantly lower expression of *SHH* and *IHH* (*Figure 1—figure supplement 1E*). Importantly, given that stromal cells also express GLI2, we confirmed that high GLI2 status was independent of stromal cell content. Of the 51 PDA tumors characterized as *GLI2* high, a relatively even distribution between high purity (>33% tumor/stromal cell content; n = 23) and low purity (<33% tumor/stromal cell content; n = 27) samples was observed (see Materials and methods for details), indicating that *GLI2* expression in stromal cells is an unlikely confounding factor in determining tumor cell GLI2 status in this patient cohort. These findings indicate that PDA cell lines and tumors segregate into GLI$^{hi}$/Hh$^{lo}$ and GLI$^{lo}$/Hh$^{hi}$ subgroups, independent of tumor cell purity, further supporting an inverse relationship between expression of Hh ligands and GLI transcription factors in PDA. Notably, high expression levels of *GLI2* in primary PDA tumors correlated with shortened overall and disease-free survival (*Figure 1D*). In contrast, high SHH expression levels were associated with longer disease-free survival (*Figure 1E*). GLI2 or SHH high versus low expression level was not correlated with clinical variables such as sex, T stage, N stage, or M stage, although as expected from our data there was a correlation between high SHH and low GLI2 expression and vice-versa (*Figure 1—figure supplement 1F,G*). A multivariate analysis of the hazard attributable to SHH or GLI2 status demonstrated that no other variables explained the relationship between SHH or GLI2 and overall or progression-free survival (*Figure 1—figure supplement 2*). Collectively, our data point to an unexpected dichotomy among PDA tumors and cell lines with respect to Hh pathway circuitry, with high GLI transcriptional activity dissociated from canonical ligand-dependent signaling and associated with worse patient outcomes. Given these findings, we sought to determine the functions of GLI2 – the main transcriptional activator amongst the GLI family of proteins (*Hui and Angers, 2011*), - in PDA and examine how its increased activity may promote a more aggressive tumor phenotype.

## GLI expression and activity correlates with a mesenchymal cell state and the basal-like subtype of PDA

We sought to determine the relationship between GLI expression and EMT in PDA, given the role of this program in cancer aggressiveness. Reexamination of our panel of 14 PDA cell lines by immunoblot (*Figure 2A*) and qRT-PCR (*Figure 1—figure supplement 1H*) for mesenchymal markers - ZEB1, ZEB2, Vimentin (VIM), SNAI1, SNAI2, N-Cadherin (CDH2) - and epithelial markers – E-Cadherin (CDH1), Epithelial splicing regulatory protein 1 and 2 (ESRP1, ESRP2) – enabled classification into EMT low and EMT high groups. Notably, whereas none of the GLI$^{lo}$/Hh$^{hi}$ cell lines showed an EMT signature, most of the GLI$^{hi}$/Hh$^{lo}$ cell lines were in the EMT high group, with the exceptions, HupT3 and Panc0327, displaying an intermediate phenotype (*Figure 2A*; *Figure 1B,C*). To assess whether GLI activity correlates with EMT status more broadly, we used a published EMT score generated from a meta-analysis of 18 independent gene expression studies of EMT (*Gröger et al., 2012*). This signature assigned 38 human PDA cell lines as EMT high or low (*Gröger et al., 2012*; *Viswanathan et al., 2017*), and we found that GLI activity (see *Figure 1B*) correlated positively with EMT in our PDA cell line panel (*Figure 2B*), while SHH expression correlated negatively (*Figure 2C*). Intermediate cell lines, Panc0327 and HupT3 clustered together with the mesenchymal and epithelial-like cohort, respectively (*Figure 2B*). Together these findings indicate that GLI$^{hi}$ status correlates with a mesenchymal cell state in PDA.

EMT and poorly differentiated histopathology are features of basal-like PDA, prompting us to examine the relationship between GLI/Hh and subtype identity. To assay classical and basal-like status in PDA cell lines, we used concordant gene signatures based on the Collisson, Moffitt and Bailey studies (*Collisson et al., 2011*; *Moffitt et al., 2015*; *Bailey et al., 2016*; *Cancer Genome Atlas Research Network, 2017*) (See Materials and methods). As expected, PDA cell lines with elevated levels of epithelial marker expression (GLI$^{lo}$/Hh$^{hi}$ lines) closely correlated with enrichment of the classical gene program (*Collisson et al., 2011*; *Aung et al., 2018*) (*Figure 2A* and *Figure 2D*). We next analyzed GLI status in 149 TCGA PDA samples pre-classified as classical or basal-like using the Collisson, Moffitt and Bailey signatures (*Cancer Genome Atlas Research Network, 2017*). Strikingly, *GLI2* mRNA expression was enriched in the basal-like tumors, (*Figure 2E* left), whereas high *SHH* and *IHH* expression correlated with the classical subtype (*Figure 2E* middle, right). Collectively these

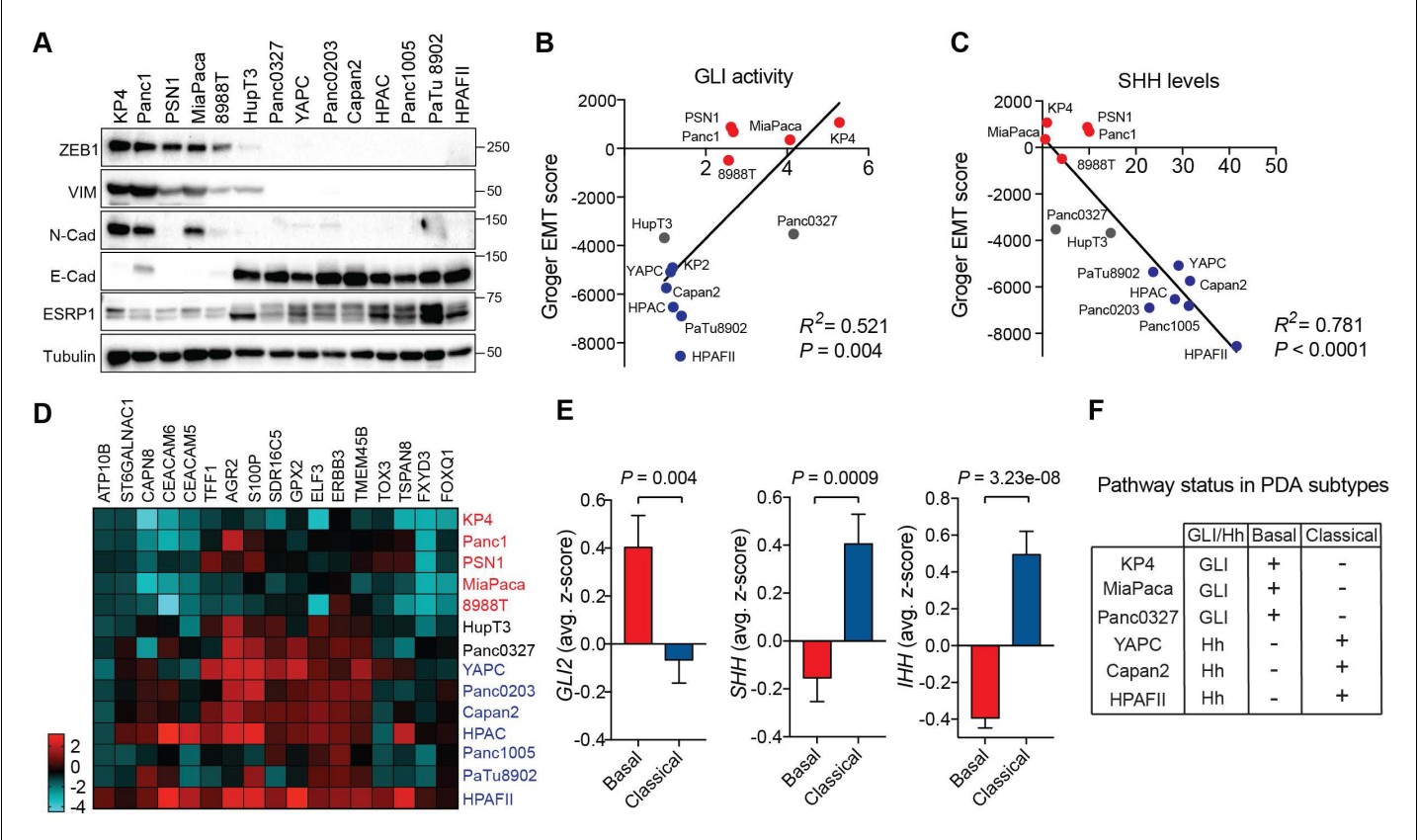

**Figure 2.** GLI expression and activity correlates with EMT and the basal-like subtype of PDA. (A) Human PDA cell lines segregate into two groups based on protein expression of EMT associated markers as indicated. Mesenchymal markers (ZEB1, VIM, N-Cadherin), epithelial markers (E-Cadherin, ESRP1). (B,C) GLI activity (B) as measured by GLI luciferase assay and SHH protein levels (see *Figure 1*) (C) correlated to EMT score. Linear regression line is shown in red along with corresponding statistics. Epithelial cell lines are indicated in blue, and mesenchymal cell lines are indicated in red. Intermediate lines are indicated in grey. (D) mRNA expression of the classical subtype gene set (see Materials and methods) across 14 human PDA cell lines. Epithelial cell lines (blue) express higher levels of the classical gene program. (E) Expression of Hh pathway components correlates with PDA subtypes. *GLI2* mRNA expression is higher in TCGA PDA samples (n = 149) classified as basal-like (n = 65), while *SHH* and *IHH* expression is higher in samples classified as classical (n = 84). Bar represents average normalized z-score. (F) Table outlining the relationship between GLI and Hh expression to PDA subtypes in PDA human cell lines.

DOI: https://doi.org/10.7554/eLife.45313.006

The following source data is available for figure 2:

**Source data 1.** GLI expression and activity correlates with EMT and the basal-like subtype of PDA.
DOI: https://doi.org/10.7554/eLife.45313.007

data demonstrate that GLI-high status is a hallmark of the basal-like, EMT-high state that portends poor patient prognosis (*Figure 2F*).

## GLI2 induction is sufficient to drive a classical to basal-like subtype switch in PDA cells

To determine whether GLI proteins have functional roles in driving subtype specification, we engineered classical subtype PDA cell lines (YAPC, HPAFII) to stably overexpress GLI2 (*Figure 3A*; *Figure 3—figure supplement 1A*). Luciferase assays revealed that ectopic GLI2 expression in YAPC cells resulted in comparable GLI transcriptional activity to what is observed in the basal-like cell lines (*Figure 3—figure supplement 1B* vs *Figure 1B*). Strikingly, GLI2 overexpression strongly induced the mesenchymal markers ZEB1, VIM and basal-like marker KRT14 in YAPC cells and led to downregulation of epithelial markers E-Cadherin and ESRP1, and the transcription factor GATA6 - a putative regulator of the classical subtype of PDA (*Collisson et al., 2011*; *Martinelli et al., 2017*) and SHH in

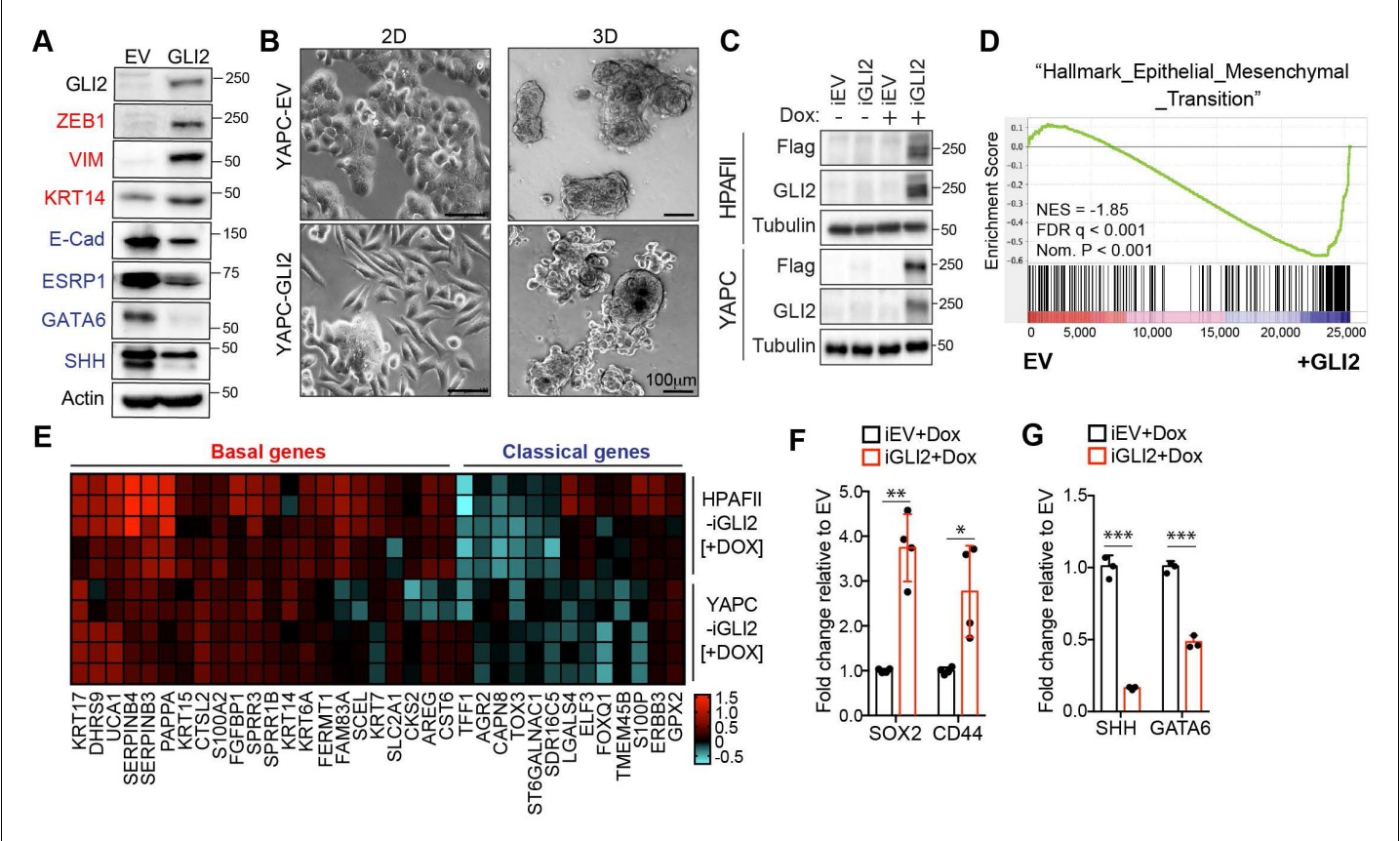

**Figure 3.** GLI2 is sufficient to drive basal-like subtype switching. (**A**) Immunoblot shows expression of the indicated proteins in YAPC cells stably expressing empty vector (EV) or Flag-tagged GLI2 (GLI2). (**B**) Images show the effect of stable expression of Flag-tagged GLI2 in YAPC cells on morphology when grown in 2D (left) and as 3D cultures (right). Note the switch to a mesenchymal phenotype in the GLI2-expressing cells. Scale bars, 100 μm. (**C**) Immunoblot shows GLI2 induction in HPAFII- and YAPC-iGLI2 cells following treatment with Dox. (**D**) Gene set enrichment analysis (GSEA) of RNA-sequencing data shows significant enrichment of the 'Hallmarks_Epithelial_Mesenchymal_Transition' gene set in YAPC-iGLI2 cells treated with 1 μg/ml Dox for 6 days to induce expression of GLI2, compared to identically treated YAPC-iEV control cells, (n = 3). (**E**) Heatmap shows expression of basal-like (red) and classical (blue) subtype associated genes determined by qRT-PCR following 1 μg/ml Dox treatment of replicate HPAFII-iGLI2 (top) and YAPC-iGLI2 (bottom) cells for 3 days. Values from n = 5 replicates per cell are normalized to gene expression in identically treated iEV control cells and log10 transformed. (**F,G**) Graph shows the effect of GLI2 induction in YAPC-iGLI2 cells treated with 1 μg/mL Dox for 3 days on *SOX2* and *CD44* (**F**) and *SHH* and *GATA6* (**G**) mRNA expression, displayed as fold change normalized to identically treated iEV control cells, (data represent n = 3 experiments). p-Values were calculated by two-tailed unpaired *t* test. *p<0.05; **p<0.01; ***p<0.001.

DOI: https://doi.org/10.7554/eLife.45313.008

The following source data and figure supplement are available for figure 3:

**Source data 1.** GLI proteins promote the basal-like phenotype.

DOI: https://doi.org/10.7554/eLife.45313.010

**Figure supplement 1.** GLI proteins promote the basal-like phenotype.

DOI: https://doi.org/10.7554/eLife.45313.009

YAPC and HPAFII cells (*Figure 3A*; *Figure 3—figure supplement 1A*). Consistent with a switch toward a mesenchymal-like cell state, YAPC- and HPAFII-GLI2 cells showed a significant change in cell morphology compared to control cells expressing empty vector (EV) (*Figure 3B*; *Figure 3—figure supplement 1C*), characterized by a more elongated, less compact morphology when grown as monolayer cultures or as 3D matrigel cultures (*Figure 3B*). Together, these data indicate that GLI2 can promote an EMT-like switch in classical PDA cells.

We next engineered YAPC and HPAFII cells to express GLI2 in a doxycycline (Dox)-inducible manner (iGLI2 cells), to study the downstream gene programs controlled by GLI2 with further resolution (*Figure 3C*). We performed RNA-seq analysis of YAPC-iGLI2 cells compared to EV control cells

(YAPC-iEV) following 6 days of Dox treatment. Gene set enrichment analysis (GSEA) using the Cancer Hallmarks database indicated that 'Epithelial_Mesenchymal_Transition' was the most statistically significant program activated by GLI2 (*Figure 3D*). We further validated that Dox-inducible activation of GLI2 in YAPC cells led to an increase in EMT markers, consistent with our results with stable GLI2 over-expression (*Figure 3—figure supplement 1D*). In keeping with the relationship between EMT and the basal-like subtype, qPCR analysis indicated that Dox treatment of iGLI2 cells for 3 days resulted in a significant increase in basal-like subtype genes and a corresponding decrease in a subset of classical genes in both YAPC- and HPAFII-iGLI2 cells (*Figure 3E*). In addition, the expression levels of stemness associated markers *SOX2* and *CD44* increased (*Figure 3F*), while *SHH* and *GATA6* showed a significant decrease upon GLI2 induction and basal-like subtype switching in YAPC cells (*Figure 3G*). Similarly, forced expression of a constitutively active GLI2 lacking the N-terminal repressor domain (*Pasca di Magliano et al., 2006*) (ΔN-GLI2) also lead to a decrease in *SHH* and *GATA6* levels in YAPC cells (*Figure 3—figure supplement 1E*). Thus, GLI2-mediated conversion from a classical to a basal-like state also incorporates loss of SHH, which is in support of our observed inverse correlation between GLI proteins and Hh ligand expression in cell lines and patient tumors.

## GLI2 is required for maintenance of the basal-like state

To test whether GLI proteins are not only sufficient but also necessary to sustain the basal-like phenotype, we suppressed GLI2 levels via siRNA or shRNA-mediated knockdown or CRISPR-Cas9 mediated knockout in basal-like cell lines, KP4 and Panc0327. Loss of GLI2 led to a decrease in basal-like markers (KRT5, KRT14) and SOX2 and induction of epithelial markers (*Figure 4A*; *Figure 4—figure supplement 1A*). Accordingly, expression of the basal-like signature and EMT-associated genes were also significantly reduced following GLI2 knockdown (KP4) (*Figure 4B,C*) or knockout (Panc0327) (*Figure 4—figure supplement 1B,C*).

Similarly, GLI2 knockout in KP4 and Panc0327 cells (GLI2$^{KO}$) resulted in a clear switch towards a more epithelial like morphology in 2D monolayer culture (*Figure 4D*) and 3D matrigel growth (KP4; *Figure 4—figure supplement 1D*). Moreover, immunofluorescence staining followingGLI2 knockdown or GLI2$^{KO}$ in KP4 and Panc0327 cells, respectively, showed a prominent nuclear relocalization of ESRP1 relative to Cas9 control cells, where ESRP1 was predominantly cytoplasmic (*Figure 4E*; *Figure 4—figure supplement 1E*). We next determined whether loss of the basal-like state compromises tumor growth. KP4 GLI2$^{KO}$ cells showed a modest difference in in vitro growth rate (*Figure 4F*); however, GLI2$^{KO}$ tumor xenografts displayed significantly reduced in vivo growth relative to Cas9 control tumors (*Figure 4G*). Importantly, slow growing KP4 GLI2$^{KO}$ tumors displayed reduced expression of the basal-like marker S100A2 (*Figure 4H*, top row) and increased expression of the classical marker GATA6 (*Figure 4H*, bottom row), suggesting that GLI2-dependent maintenance of a basal-like state is important for facilitating rapid in vivo tumor growth.

Collectively, these data indicate that GLI2 is both necessary and sufficient to drive a common program that couples EMT and the basal-like subtype of PDA and that a significant level of cellular plasticity exists within PDA molecular subtypes

## GLI2 mediates basal-like subtype switching in response to oncogenic KRAS$^{G12D}$ ablation

Recent studies have shown that PDA tumor cells can survive in the absence of oncogenic KRAS (referred to hereafter as KRAS*) signaling by activating alternative mechanisms that maintain their growth (*Singh et al., 2009*; *Kapoor et al., 2014*; *Kemper et al., 2014*; *Shao et al., 2014*). These circuits often restore critical functions of KRAS*, such as tumor cell proliferation and evasion of apoptosis, thereby enabling cancer cells to escape KRAS* withdrawal. Moreover, basal-like PDA cell lines have been shown to harbor reduced dependency on KRAS* for growth (*Singh et al., 2009*). Therefore, we hypothesized that GLI2-mediated basal-like subtype switching of classical PDA cells could likewise obviate KRAS*-dependency. First, we measured GLI1 and GLI2 expression levels in response to oncogenic Kras* extinction in cells derived from a murine PDA model driven by a Dox-inducible Kras* allele [termed the iKRAS model (*Ying et al., 2012*). In this system, removal of Dox from the culture media led to a 50% reduction in *Kras* mRNA expression within 24 hr and complete extinction by 72 hr (*Figure 5—figure supplement 1A,B*). Interestingly, a reciprocal increase in *Gli2* and *Gli1*

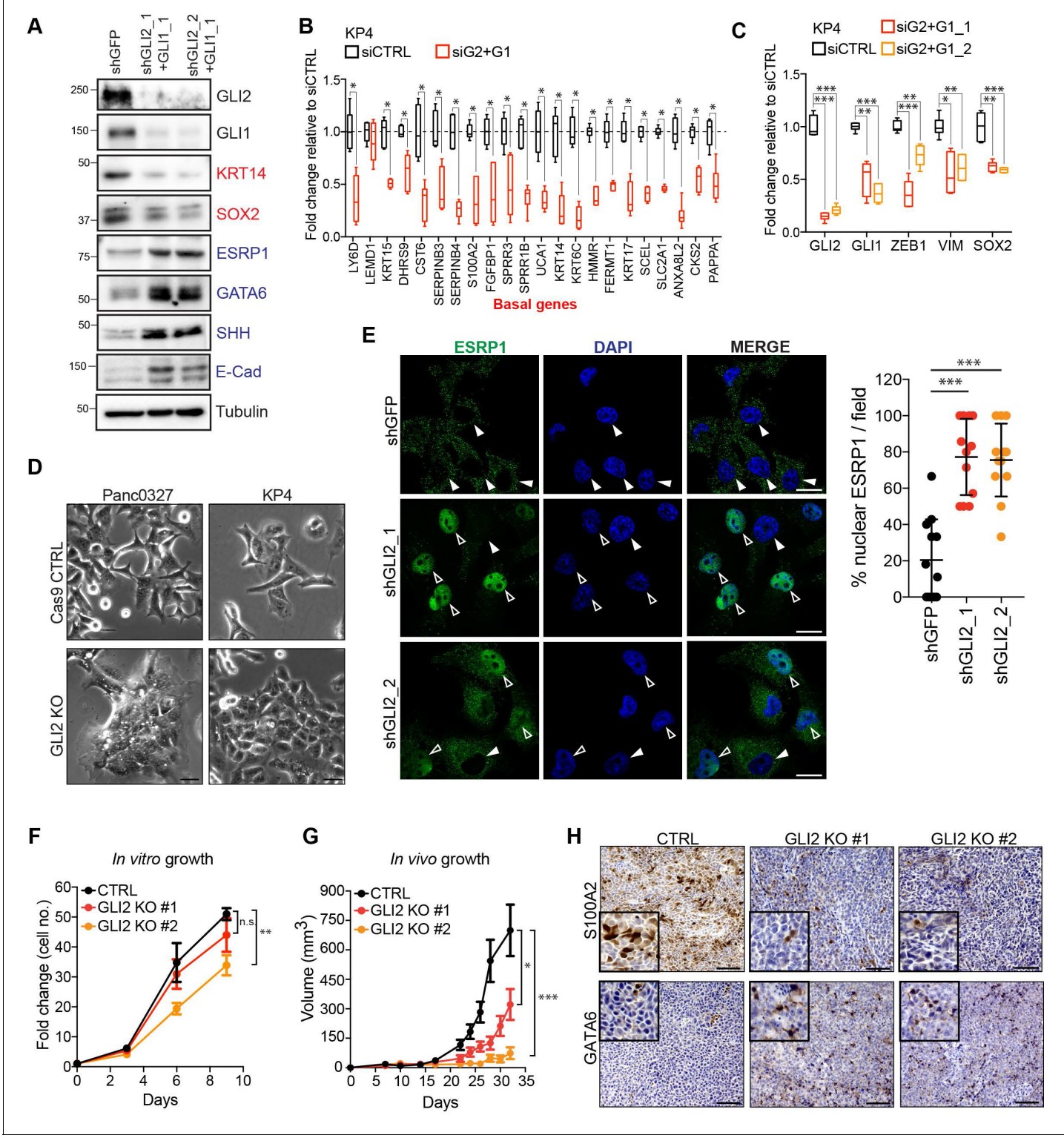

**Figure 4.** GLI2 is required to maintain the basal-like state in PDA. (**A**) Immunoblot shows the effect of shRNA-mediated GLI2 and GLI1 combined knockdown on expression of the indicated proteins in KP4 cells. (**B,C**) Graph shows the effect of siRNA mediated GLI2 and GLI1 combined knockdown on EMT associated gene expression (**B**) and basal-like subtype gene expression (**C**) determined by qRT-PCR, displayed as fold change normalized to a scrambled siRNA control in KP4 cells (n = 2). (**D**) Brightfield images of Panc0327 and KP4 cells grown in 2D monolayer show loss of a mesenchymal phenotype upon GLI2 knockout (GLI2 KO) compared to Cas9 control cells (CTRL). Scale bar, 100 μm. (**E**) Immunofluorescence staining (left) and quantification (right) of percentage ESRP1 nuclear localization in KP4 control (shGFP) and shRNA mediated GLI2 knockdown cells. Data represent 10

*Figure 4 continued on next page*

*Figure 4 continued*

fields from two independent experiments. Solid arrowheads indicate absence of nuclear ESRP1; open arrowheads indicate nuclear ESRP1. Scale bar, 50 µm. (**F**) Quantification of in vitro growth rate of KP4 Cas9 CTRL and GLI2 KO cells. (**G**) Quantification of in vivo growth of KP4 CTRL and GLI2 KO subcutaneous xenografts. Error bars represent s.e.m. (**H**) Immunohistochemistry of KP4 xenografts show downregulation of basal-like marker S100A2 (top) and upregulation of GATA6 (bottom) in GLI2 KO cells compared to CTRL, consistent with loss of the basal-like state. Scale bars, 200 µm. p-Values were calculated by two-tailed unpaired *t* test. n.s. = not significant; *p<0.05; **p<0.01; ***p<0.001.

DOI: https://doi.org/10.7554/eLife.45313.011

The following source data and figure supplement are available for figure 4:

**Source data 1.** GLI2 is required to maintain a basal-like state in PDA.

DOI: https://doi.org/10.7554/eLife.45313.013

**Figure supplement 1.** GLI2 is required to maintain a basal-like state in PDA.

DOI: https://doi.org/10.7554/eLife.45313.012

expression was evident within 24 hr of Dox removal and persisted for at least 5 days (*Figure 5A–C*; *Figure 5—figure supplement 1A*). Moreover, there was a corresponding up-regulation of *Zeb1* and *Vim* mRNA and of basal-like subtype genes, consistent with induction of EMT and basal-like features (*Figure 5—figure supplement 1C,D*). Similarly, shRNA-mediated knockdown of *KRAS* in the human PDA cell lines YAPC and HPAFII led to a concomitant increase in GLI expression, EMT markers (*Figure 5D*; *Figure 5—figure supplement 1E*) and basal-like subtype genes (*Figure 5E*). Importantly, expression of the basal-like program was dependent on GLI2, as treatment of iKRAS4 cells with the GLI inhibitor, GANT61 (5 µM), suppressed basal-like gene induction in response to KRAS* ablation (*Figure 5F*). Thus, upon KRAS* suppression, upregulation of GLI expression is responsible for induction of the basal-like gene program.

## Acquired resistance and tumor relapse following KRAS* ablation is mediated by GLI2

To test whether GLI2 induction is required to sustain growth of tumor cells following suppression of KRAS*, we stably overexpressed Flag-tagged mouse GLI2 in iKRAS cell lines (iKRAS-GLI2) prior to Dox withdrawal (*Figure 5—figure supplement 2A*). Consistent with our findings in human PDA cell lines following ectopic expression of GLI2, murine iKRAS-GLI2 cells similarly displayed an increase in expression of basal (KRT5, KRT14) and EMT markers and SOX2 (*Figure 5—figure supplement 2B*). In the presence of Dox (KRAS* on), iKRAS-GLI2 cells had a growth advantage over control cells (*Figure 5G*, quantified in graph; '+Dox'). Upon Dox withdrawal (KRAS* off), control cells (iKRAS-EV) failed to form colonies after 6 days while iKRAS-GLI2 cells were able to form colonies within the same time frame (*Figure 5G*, quantified in graph; '- Dox'; *Figure 5—figure supplement 2C*). This phenotype was recapitulated in 3D spheroid culture, where ectopic GLI2 expression rescued growth of iKRAS cells by 2.5 fold following Dox withdrawal (*Figure 5—figure supplement 2D*). Similarly, human YAPC-GLI2 PDA cells were capable of sustained growth as 3D spheroids following shRNA-mediated knockdown of KRAS* relative to YAPC-EV cells (*Figure 5H,I*; *Figure 5—figure supplement 2E*). Thus activation of GLI2 in PDA cells accelerates the emergence of clones that grow in the absence of KRAS*.

Sustained KRAS* pathway suppression in the iKRAS model was shown to induce significant tumor regression; however, this was often followed by rapid tumor re-growth (*Kapoor et al., 2014*; *Shao et al., 2014*). Relapsed tumors retain KRAS* pathway suppression (*Figure 5J*) and a subset harbor genomic amplification of the Hippo pathway transcriptional coactivator, YAP1 (*Kapoor et al., 2014*), which we have confirmed in 3 of 7 'Escaper' tumor cells lines (*Figure 5J*; cell lines denoted as EY1-3). Interestingly, several Escaper cell lines do not harbor YAP1 amplification, suggesting that additional mechanisms may exist which mediate bypass of KRAS* extinction. Strikingly, we find that GLI2 is upregulated in 6/7 Escaper cell lines, including 4/7 lacking YAP1 induction. These results suggest that PDA cells co-opt several bypass mechanisms to circumvent KRAS* inactivation, including YAP1 and GLI2 upregulation, which may functionally compensate for KRAS* loss.

To further test the role of GLI2 in tumor relapse following KRAS* suppression, we grew iKRAS cell lines (iKRAS1 and iKRAS4) in the presence or absence of Dox for 15–18 days following shRNA mediated knockdown of *Gli2* (*Figure 5—figure supplement 2F*). In the presence of Dox, loss of GLI2 did not have a significant effect on spheroid growth compared to shGFP control cells (*Figure 5K* top

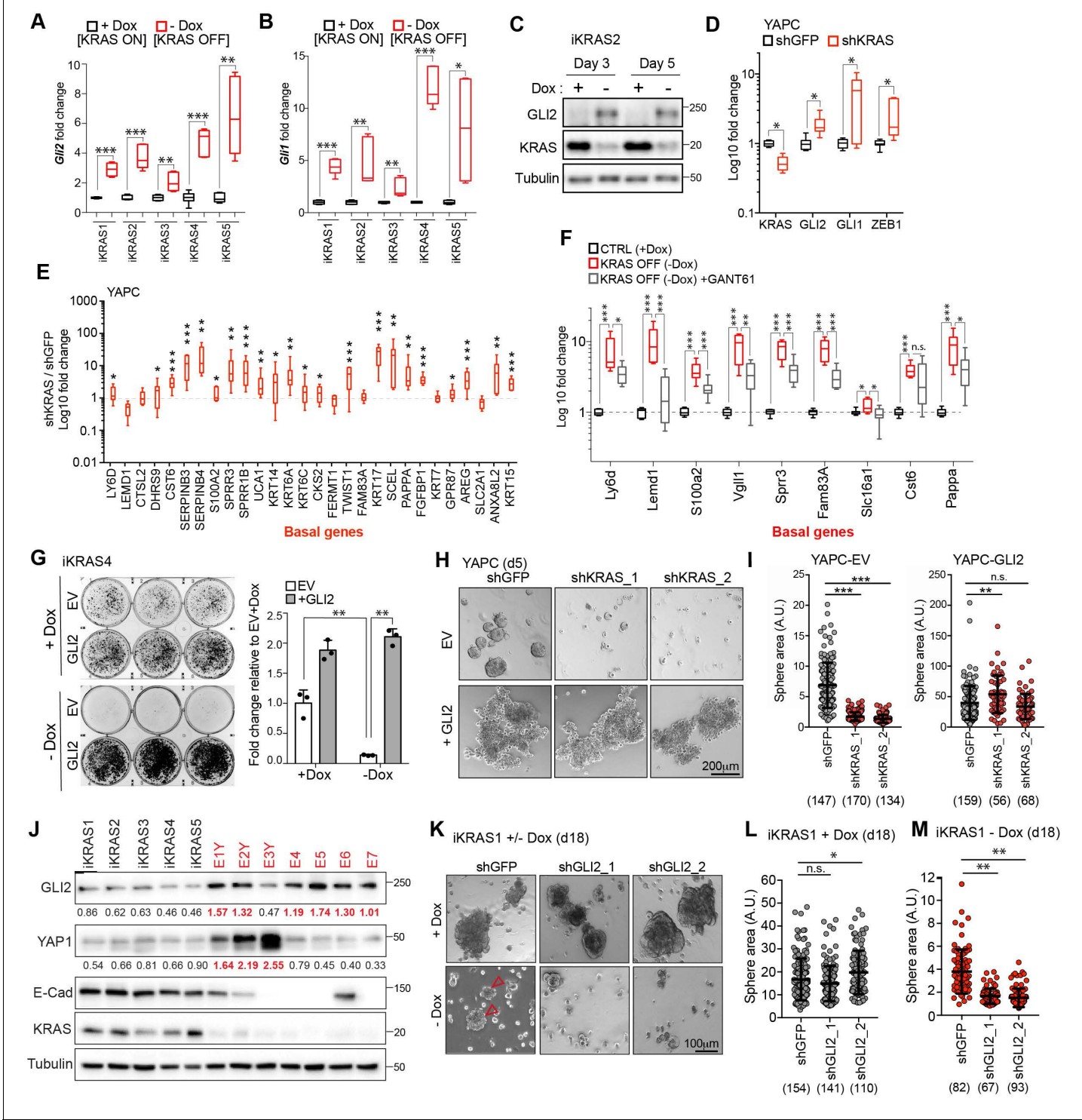

**Figure 5.** GLI2-mediated basal-like subtype switching rescues viability of PDA cells following KRAS[G12D] ablation. (**A,B**) Effect of Dox depletion for 3 days on levels of *Gli2* (**A**) and *Gli1* (**B**) mRNA as measured by qRT-PCR in the indicated iKRAS cell lines, (n = 3). (**C**) Immunoblot shows expression of GLI2 and KRAS in iKRAS2 cells + Dox and – Dox for 3 and 5 days. (**D,E**) Effect of shRNA-mediated knockdown of *KRAS* on *GLI2, GLI1, ZEB1* mRNA levels (**D**) and basal-like gene expression (**E**) as measured by qRT-PCR in YAPC cells. (n = 3). (**F**) Fold change in the basal-like gene signature in iKRAS4 cells following Dox removal (red plot) for 3 days or Dox removal in conjunction with 5 μM GANT61 treatment (grey plot). qRT-PCR measurement of the indicated genes in both conditions are normalized to the +Dox control condition (black plot) (n = 3). (**G**) Colony-forming ability of iKRAS4 cells stably expressing empty vector (EV) or Flag-tagged mGLI2 (GLI2) grown in the presence (top) or absence (bottom) of Dox for 6 days, followed by staining with crystal violet. Graph shows the fold change in growth relative to the EV +Dox setting. Results shown are representative of n = 3 experiments. (**H,I**)

*Figure 5 continued on next page*

Figure 5 continued

Images (H) show the effect of stable expression of EV (top) or Flag tagged GLI2 (bottom) on YAPC sphere formation 5 days post shRNA-mediated knockdown of *KRAS* compared to control cells (shGFP). Quantification of sphere area for each condition is shown (I). Number of spheres measured per condition is indicated in parenthesis. Scale bar, 200 μm. (J) Immunoblots show expression of the indicated proteins in iKRAS (black) and Escaper (red) cell lines. Numerical values indicate levels of GLI2 and YAP1 normalized to Tubulin. Escaper cell lines denoted with 'Y' harbor genomic amplification of *Yap1*. (K–M) Images (K) show the effect of shRNA-mediated knockdown of *Gli2* using two separate hairpins on iKRAS1 sphere formation when grown in the presence (top) or absence (bottom) of Dox for 18 days. Quantification of sphere area in the presence (L) and absence (M) of Dox is shown. Number of spheres measured per condition is indicated in parenthesis. Arrowheads indicate growth of KRAS$^{G12D}$ independent spheroids. Scale bar, 100 μm. p-Values were calculated by two-tailed unpaired *t* test. n.s. = not significant, *p<0.05; **p<0.01; ***p<0.001.
DOI: https://doi.org/10.7554/eLife.45313.014

The following source data and figure supplements are available for figure 5:

Source data 1. GLI2-mediated basal-like subtype switching in response to KRAS$^{G12D}$ ablation.
DOI: https://doi.org/10.7554/eLife.45313.017
Figure supplement 1. GLI2 is upregulated in response to KRAS$^{G12D}$ ablation.
DOI: https://doi.org/10.7554/eLife.45313.015
Figure supplement 2. GLI2 is necessary and sufficient to promote bypass of Kras$^{G12D}$-mediated oncogene addiction.
DOI: https://doi.org/10.7554/eLife.45313.016

row, L; *Figure 5—figure supplement 2G* top row, H). However, the eventual emergence of resistant colonies in the absence of Dox was suppressed in shGLI2 cells compared with shGFP control (*Figure 5K* bottom row, M arrowheads; *Figure 5—figure supplement 2G*, bottom row, arrowheads and I). Thus, GLI2-mediated basal-like subtype switching can rescue viability upon KRAS* suppression, whereas its loss exacerbates the growth deficits caused by KRAS* loss.

## The secreted ligand osteopontin (OPN) promotes basal-like subtype conversion downstream of GLI2

We next interrogated candidate downstream GLI2 targets that may drive subtype interconversion. Among the most upregulated transcripts expressed in YAPC-iGLI2 cells following GLI2 induction were genes encoding secreted factors including Secreted Phosphoprotein 1 (SPP1) encoding Osteopontin (OPN), fibroblast growth factor 19 (FGF19) and SPOCK2 [SPARC (Osteonectin)] and the basal-like gene S100A2 (*Figure 6A*). OPN is a secreted glycosylated phosphoprotein classified as a member of the 'small integrin-binding ligand N-linked glycoproteins' (SIBLINGs) and has been associated with tumorigenesis and metastasis of several tumor types (*Yoon et al., 2002*; *Rangaswami et al., 2006*; *Das et al., 2009*; *Pietras et al., 2014*; *Ahmed et al., 2016*; *Zhao et al., 2018*). We confirmed upregulation of secreted OPN in YAPC-iGLI2 (*Figure 6—figure supplement 1A*) and in a second cell line, HPAFII-iGLI2 (*Figure 6B,C*) 3 days post-induction of GLI2. Similarly, ectopic expression of ΔN-GLI2 in YAPC cells also led to an increase in *SPP1* expression (*Figure 6— figure supplement 1B*). Consistent with direct regulation of *SPP1* transcription, chromatin immunoprecipitation of GLI2 showed binding to a GLI consensus element found upstream of the human *SPP1* transcriptional start site (*Figure 6D*) (*Kijewska et al., 2017*).

Expression of *SPP1* is significantly increased in YAPC cells during basal-like subtype switching in response to KRAS* suppression (*Figure 6E*), suggesting that OPN induction is a hallmark of the basal-like cell state in PDA. Accordingly, we found that GLI2-high basal-like cell lines expressed the highest levels of *SPP1* mRNA (*Figure 6—figure supplement 1C*) and OPN protein levels in addition to higher levels of OPN receptors, CD44 and integrin β3, as well as integrin α5 (*Figure 6F*) (*Weber et al., 1996*; *Rangaswami et al., 2006*; *Zhao et al., 2018*), compared to classical subtype cells. Moreover, knockdown of GLI2 alone or in combination with GLI1 led to a significant decrease in *SPP1* transcript (*Figure 6G,H*; *Figure 6—figure supplement 1D*) and secreted protein (*Figure 6— figure supplement 1E*), establishing GLI transcription factors as endogenous regulators of *SPP1* expression.

We next tested whether treatment of classical subtype PDA cells with exogenous OPN is sufficient to induce a basal-like subtype switch. Exposure of Capan2 cells to recombinant OPN led to a significant increase in expression of basal-like genes, and a decrease in a subset of classical genes (*Figure 6I*). A similar induction of basal-like genes was also observed in YAPC cells treated with exogenous OPN (*Figure 6—figure supplement 1F*). To assess whether OPN promotes a basal-like

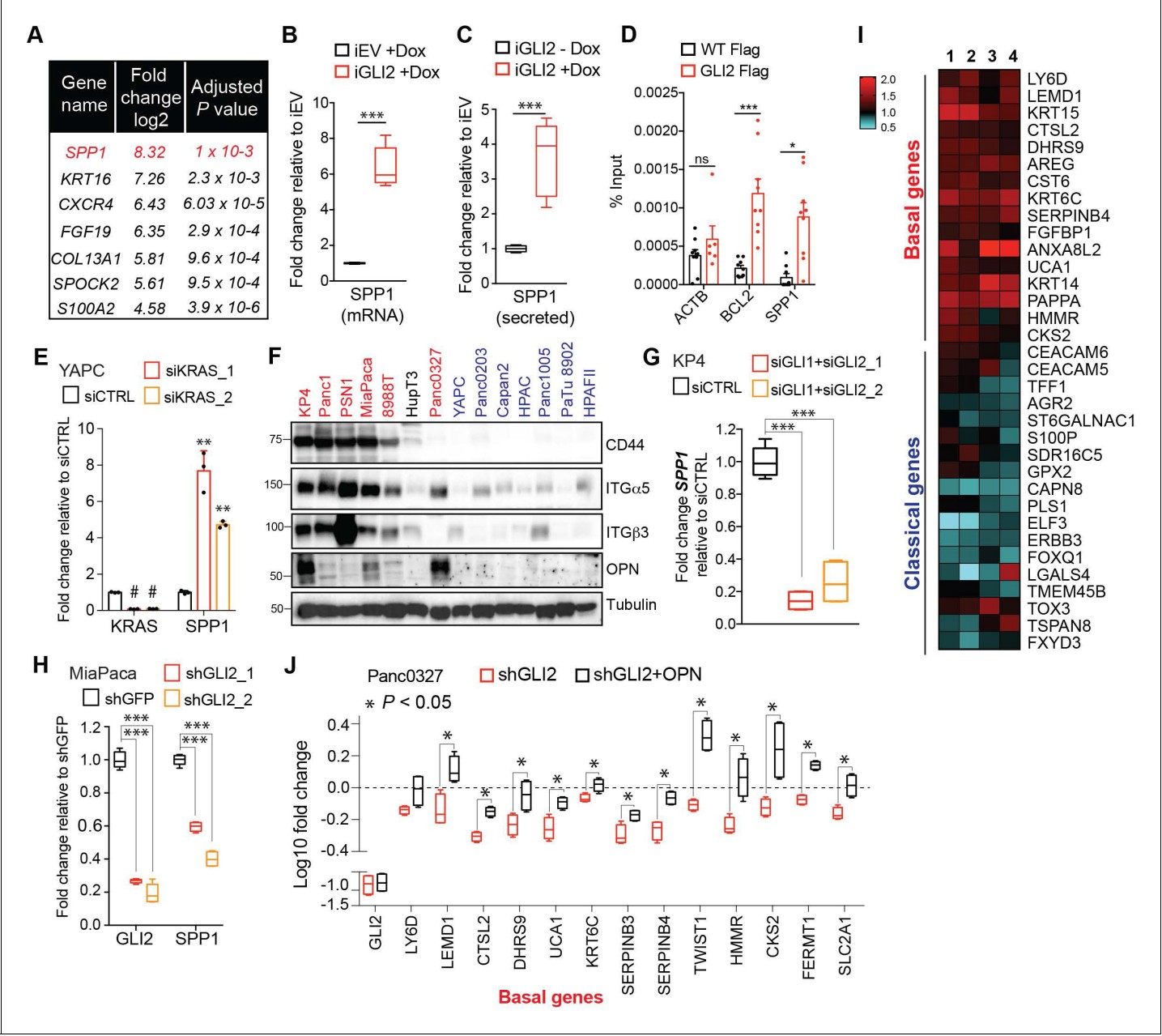

**Figure 6.** OPN is a downstream effector of GLI2 that promotes a basal-like subtype switch. (**A**) Top altered genes identified via RNA-sequencing analysis of YAPC-iGLI2 cells + Dox relative to YAPC-iEV +Dox, (n = 3). (**B,C**) Graph shows the relative fold change in human *SPP1* mRNA (**B**) and secreted protein (OPN) in the conditioned media as measured by ELISA (**C**) of HPAFII-iGLI2 cells treated with 1 µg/mL Dox for 3 days normalized to identically treated HPAFII-iEV. (**D**) Chromatin immunoprecipitation of Flag-tagged GLI2 in 293 T cells shows enrichment of binding to the *BCL2* and *SPP1* promoter relative to WT control cells (n = 3). Error bars indicate s.e.m. (**E**) Effect of siRNA-mediated knockdown of *KRAS* in YAPC cells on *SPP1* mRNA expression as measured by qRT-PCR. (**F**) Western blot of OPN receptors, CD44, integrin α5 and integrin β3 in basal-like (red) and classical (blue) PDA cell lines. (**G,H**) Effect of *GLI2* knockdown in KP4 (**G**) and MiaPaca (**H**) cells, on mRNA expression of the indicated genes. (**I**) Heatmap shows the effect of treating Capan2 cells with 1 µg/mL recombinant human OPN for 3 days on expression of the basal-like and classical subtype genes as measured by qRT-PCR (n = 4). Values are normalized to no treatment and log2 transformed. (**J**) Effect of shRNA-mediated knockdown of *GLI2* alone (red) or in conjunction with 1 µg/mL OPN treatment (black) on expression of basal genes in Panc0327 cells. Values are normalized to no treatment and log10 transformed. p-Values were calculated by two-tailed unpaired *t* test. *p<0.05; **p<0.01; ***p<0.001.

DOI: https://doi.org/10.7554/eLife.45313.018

The following source data and figure supplement are available for figure 6:

**Source data 1.** GLI2-mediated basal-like subtype switching in response to KRAS[G12D] ablation.

DOI: https://doi.org/10.7554/eLife.45313.020

*Figure 6 continued on next page*

*Figure 6 continued*

**Figure supplement 1.** OPN is a downstream effector of GLI2 and promotes the basal-like subtype of PDA.
DOI: https://doi.org/10.7554/eLife.45313.019

program downstream of GLI2, we determined the ability of exogenous OPN to rescue basal gene expression upon GLI2 knockdown in basal-like cells. shRNA-mediated GLI2 knockdown in Panc0327 cells led to a decrease in 13 basal-like genes, of which 12 were significantly rescued following co-treatment with recombinant OPN (*Figure 6J*).

## Loss of OPN in basal-like cells suppresses in vivo tumor growth

To test the role of the GLI-OPN axis in tumorigenesis, we first generated an OPN-deleted KP4 line via CRISPR-mediated gene editing (*Figure 7A*). OPN-deleted KP4 cells displayed a decrease in basal-like markers KRT14 (*Figure 7B*) and a concomitant increase in classical-associated markers, E-cadherin and GATA6 (*Figure 7B,C*), along with a switch toward a more epithelial like morphology on monolayer culture (*Figure 7D*).

Similar to GLI2$^{KO}$ cells, SPP1$^{KO}$ cells did not show a significant growth defect in vitro relative to Cas9 controls cells (*Figure 7E*). However, SPP1$^{KO}$ cells displayed a dramatic defect in growth as tumor xenografts in mice (*Figure 7F,G*). Similarly, SPP1$^{KO}$ cells were significantly impaired in their ability to form metastatic tumors in the lungs of recipient mice following tail vein injection, whereas Cas9 control cells gave rise to widespread metastatic disease (*Figure 7H,I*). Accordingly, the reduced overall tumor burden of mice injected with SPP1$^{KO}$ cells led to their prolonged survival relative to the control cohort (*Figure 7J*). These data indicate that OPN, downstream of GLI2, is a major growth regulator of basal-like tumors. Consistent with this notion, high expression levels of *SPP1* mRNA in basal-like tumors correlated with shortened overall survival of PDA patients (*Figure 7K*).

Collectively, our findings support a key role for a GLI-OPN axis in promoting and maintaining a basal-like phenotype that is key for PDA tumorigenesis, and highlight the importance of GLI2-dependent, dynamic inter-conversion between subtypes in enabling adaptation to cellular stress, such as following loss of oncogenic KRAS (*Figure 7L*).

## Discussion

While the clinical relevance of PDA subtypes is becoming increasingly appreciated, a mechanistic understanding of the molecular underpinnings of PDA subtype identity is lacking. Our findings establish a broad equivalence between basal-like and EMT-high, versus classical and EMT-low states of PDA. Moreover, we have identified a novel role for the GLI2 transcription factor as a central driver for promoting and maintaining the basal-like state in PDA. We found that high GLI2 status independently predicts shortened survival of PDA patients and correlates with the aggressive basal-like subtype of PDA. In contrast, high levels of the Hh ligands correlate with longer patient survival and are associated with the classical subtype of PDA. These findings highlight a remarkable rewiring of the Hh signaling pathway in PDA, whereby expression of Hh ligands and GLI proteins trend toward mutual exclusivity and instead correlate with the classical and basal-like subtype of PDA, respectively (*Figure 7L*).

Importantly, GLI2 is required for maintaining the basal-like state as GLI2 knockdown in tumor derived human and mouse PDA cell lines suppresses the basal-like program in vitro, attenuating expression of mesenchymal and stem cell markers. Conversely, forced expression of GLI2 in classical subtype cells was sufficient to induce a switch to a basal-like phenotype. Thus, GLI2 has critical functions in promoting and maintaining the basal-like phenotype and conferring enhanced plasticity to PDA cells by enabling interconversion between subtype states.

Mechanistically, our findings indicate that GLI2 induced a broad transcriptional program in PDA with a particular enrichment in EMT genes. Within this program, we identified several secreted proteins including OPN, which are induced following forced GLI2 expression in classical subtype cells. Accordingly, basal-like PDA cells express higher levels of OPN and several of its known receptors, including CD44 and integrin β3. Our studies also suggest that OPN, like GLI2, is a key determinant of cellular reprogramming. As with GLI2, CRIPSR-mediated knockout of OPN in basal-like cells led to a failure to maintain the basal-like state and induced a classical subtype switch. Importantly, loss

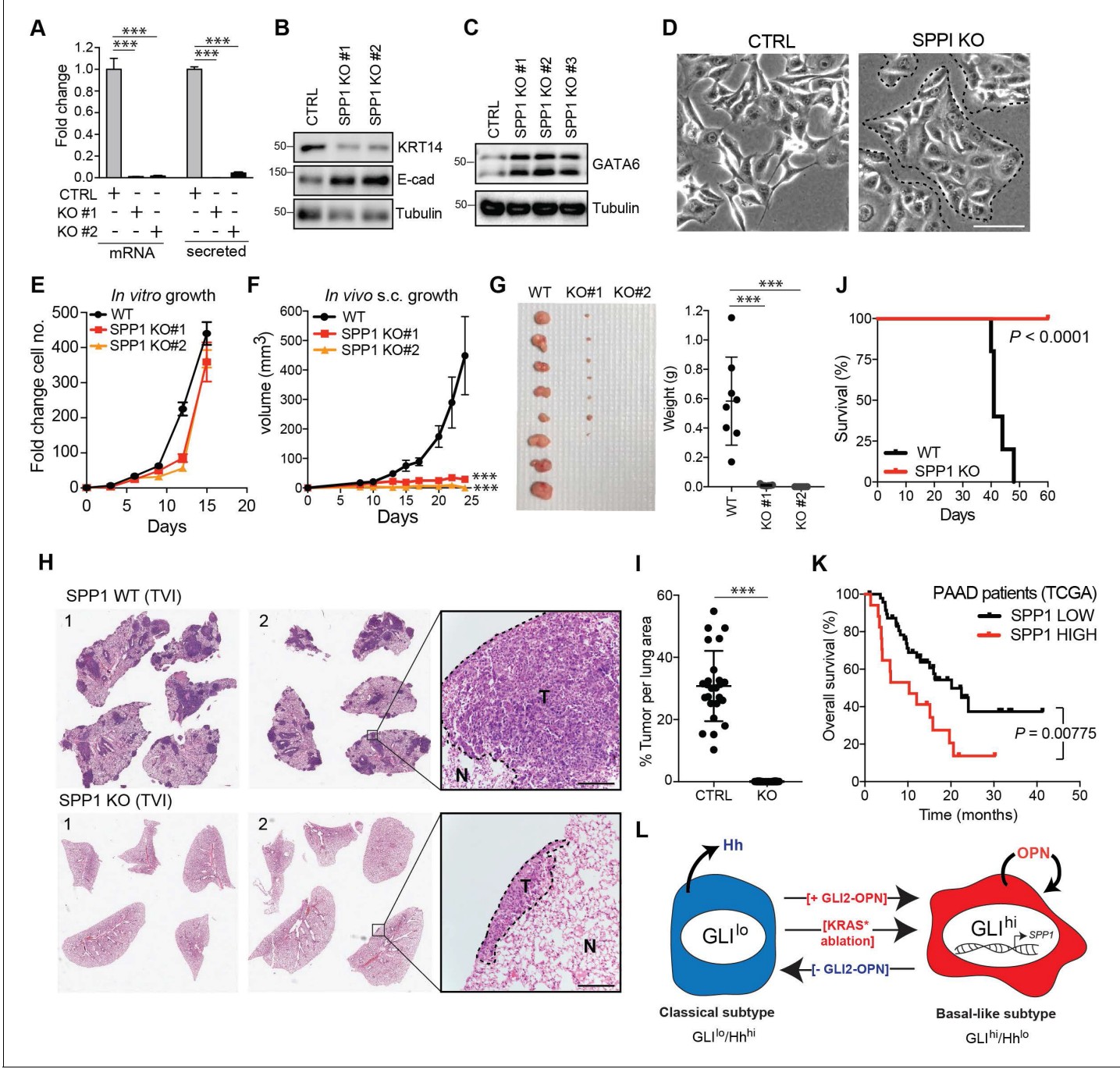

**Figure 7.** OPN loss impairs growth of basal-like PDA. (**A**) CRISPR-Cas9-mediated knockout of SPP1 in KP4 cells leads to a decrease in mRNA expression and secreted Osteopontin in two independent clones. (**B,C**) Immunoblot of KRT14 and E-cadherin (**B**) and GATA6 (**C**) levels following KO of *SPP1* in KP4 cells. (**D**) Brightfield images of KP4 cell morphology when grown in 2D monolayer following KO of *SPP1*. Note the switch to a more epithelial phenotype in the KO cells. Scale bars, 100 μm. (**E**) In vitro growth rate of KP4 control versus *SPP1* KO cell lines. Error bars represent s.e.m. (**F**) Growth rate of control and *SPP1* KO KP4 cells following subcutaneous injection into the flank of NOD/SCID mice. Error bars represent s.e.m. (**G**) Relative size of control and *SPP1* KO KP4 xenografts resected at day 25 (left) and comparative tumor weight (graph at right). (**H**) Representative H and E stained sections of lungs at endpoint following tail vein injection (TVI) of 2 × 10⁶ KP4 control cells (WT; top) or *SPP1* KO (bottom) into NOD/SCID mice. Scale bars, 400 μm. (**I**) Quantification of metastatic tumor burden in the lungs as a percentage of total lung area per mouse (n = 5 CTRL; n = 10 SPP1 KO). (**J**) Kaplan-Meier graph depicting survival of NOD/SCID mice following TVI of KP4 WT (black) or *SPP1* KO cells (red). (**K**) High expression of *SPP1* in basal-like PDA patient tumors from TCGA predicts shorter overall survival (*SPP1* high n = 17, *SPP1* low n = 48). Data from 65 patients. (**L**) Model depicting a classical to basal-like switch mediated by a GLI2-OPN signaling axis and in response to $KRAS^{G12D}$ ablation. The classical and basal-like

*Figure 7 continued on next page*

*Figure 7 continued*

subtypes are marked by GLI$^{lo}$/Hh$^{hi}$ and GLI$^{hi}$/Hh$^{lo}$ expression, respectively. p-Value calculated by Log-rank test and by two-tailed unpaired *t* test. *p<0.05; **p<0.01; ***p<0.001.

DOI: https://doi.org/10.7554/eLife.45313.021

The following source data is available for figure 7:

**Source data 1.** OPN loss impairs growth of basal-like PDA.

DOI: https://doi.org/10.7554/eLife.45313.022

of OPN in basal-like PDA cells completely suppressed primary and metastatic tumor growth indicating that maintenance of the basal-like state is required for facilitating rapid in vivo tumor growth.

Notably, OPN has been linked to aggressive disease features including increased stemness, migration, EMT and drug resistance in several other cancer settings (*Rangaswami et al., 2006*; *Orian-Rousseau, 2010*; *Das et al., 2013*; *Wang et al., 2015*; *Zhao et al., 2018*). In the context of PDA, early studies showed that OPN was over-expressed in patient tumors and could promote the growth of PDA cells (*Koopmann et al., 2004*; *Kolb et al., 2005*). Subsequent studies measuring serum OPN have suggested that elevated levels can distinguish PDA from chronic pancreatitis and healthy control subjects (*Koopmann et al., 2004*; *Poruk et al., 2013*; *Cao et al., 2019*). Whether OPN can serve as a faithful diagnostic and/or prognostic biomarker of aggressive disease and more specifically, the basal-like subtype will be an important future direction.

Our data also shows that exogenous OPN can drive a classical to basal-like subtype switch. This finding suggests the intriguing possibility that GLI2-dependent induction of secreted factors may serve as a juxtacrine or paracrine mechanism to instruct neighboring tumor cells to establish and/or sustain a basal-like cell state. Tumor associated stromal cells such as activated macrophages (*Koopmann et al., 2004*) and pancreatic stellate cells (*Cao et al., 2019*) can also secrete OPN and therefore potentially promote a basal-like switch. Thus, the ability to locally reprogram neighboring epithelial cells to a more aggressive cell state may contribute to disease progression. In contrast to OPN, tumor derived SHH is associated with the classical subtype of PDA and was shown to restrain tumor growth (*Lee et al., 2014*; *Mathew et al., 2014*; *Rhim et al., 2014*; *Liu et al., 2016*). Recent studies in bladder cancer suggest that secreted SHH functions to activate the release of stromal derived paracrine cues that promote epithelial differentiation of tumor cells (*Shin et al., 2014*). Consistent with this finding, our data in PDA show that high SHH levels correlate with the well-differentiated classical subtype of PDA and longer disease-free survival of PDA patients. Taken together, these findings indicate opposite effects of OPN and SHH on PDA differentiation status and growth and suggest a potential hierarchy between secreted cues may help to establish subtype identity in PDA.

Recent studies have established that PDA subtypes have distinct epigenetic landscapes that underlie their specific transcriptional signatures (*Lomberk et al., 2016*; *Andricovich et al., 2018*; *Somerville et al., 2018*). Notably, a subset of basal-like/squamous PDA exhibit enhancer reprogramming mediated by TP63, consistent with the functions of this transcription factor as a master regulator of squamous differentiation. The impact of GLI2 on chromatin states and the specific interplay between GLI2 and other transcription factors and epigenetic regulators will be important to address in future studies to fully decipher the circuitry driving subtype switching in PDA.

The ability of cancer cells to adaptively interconvert between states that rely on different molecular circuits to support growth and survival provides a potential mechanism for treatment failure and tumor relapse (*Polyak and Weinberg, 2009*). Notably, a prior study has linked elevated GLI expression to resistance to therapy in patients with acute myeloid leukemia (AML) (*Zahreddine et al., 2014*). Reciprocally, switching off GLI activity in basal cell carcinoma of the skin was associated with a cell identity switch involving induction of GATA6 (*Biehs et al., 2018*). In the context of PDA, the ability to switch between PDA subtypes may likewise facilitate therapeutic resistance or bypass of oncogene addiction. Our data indicates that activation of a GLI-OPN circuit is a novel mechanism of acquired resistance to KRAS* inhibition. Upon KRAS* loss, GLI2 is induced and substitutes for KRAS* suppression by promoting a basal-like switch. A prediction based on these observations is that cells with pre-existing high GLI activity would show intrinsic resistance to KRAS* inhibition and potentially

other oncogenic drivers. As strategies to target oncogenic KRAS are currently in development, co-targeting of GLI – and/or OPN in PDA - may be necessary to achieve durable therapeutic responses.

In conclusion, we have identified a high level of intrinsic plasticity between PDA subtypes mediated by GLI proteins and uncovered a surprising and unconventional role for these transcription factors in maintenance of the basal-like state. Blocking the ability of tumor cells to dynamically switch between cell states via inhibition of GLI proteins, OPN and other master regulators of cellular plasticity (*Andricovich et al., 2018*; *Somerville et al., 2018*) will be an important future direction in combating intrinsic and acquired resistance to therapy in PDA and other cancers.

## Materials and methods

### Antibodies and reagents

Antibodies against GLI2 H-300 (sc-28674) and GAPDH (sc-32233) were purchased from Santa Cruz Biotechnology; GLI3 (AF3690) from R and D Systems; SHH (ab53281), SMO (ab38686) and S100A2 (ab109494) from Abcam; GLI2 (18989–1-AP), KRAS (12063–1-AP), Tubulin (66031–1-Ig), E-Cadherin (20874–1-AP), and ESRP1 (21045–1-AP) from Proteintech; GLI1 (2534), FLAG (2368), ZEB1 (3396), Vimentin (5741), YAP1 (4912), N-Cadherin (4061), GATA6 (5851) from Cell Signaling Technology; Keratin 14 (905304) and Keratin 5 (905504) from BioLegend; Peroxidase goat anti-rabbit (PI-1000), horse anti-goat (PI-9500), and horse anti-mouse IgG (PI-2000) antibodies from Vector Labs. Keratinocyte SFM and supplements, RPM1, DMEM, DMEM F12 (1:1), fetal bovine serum (FBS) were purchased from Corning; tet system approved FBS from Clontech. Silencer select siRNA sequences against human GLI2, GLI1 and KRAS were purchased from Ambion.

### Cell culture

Cell lines were obtained from the American Type Culture Collection (ATCC). Mouse PDA Dox-inducible iKRAS lines were gifts from Haoqiang Ying and Alec Kimmelman. Human PDA cell lines were cultured as previously described (*Perera et al., 2015*) and iKRAS mouse cell lines were cultured as previously described (*Kapoor et al., 2014*). All cell lines were confirmed negative for mycoplasma contamination on a routine basis using MycoAlert Detection Kit (LT07-418) from Lonza. Colony formation was assessed following plating of 3,000 cells per well, which were fixed with 4% paraformaldehyde and stained with 0.1% crystal violet after 6 days of growth. Cells grown in three-dimensiaonal culture were plated at 10,000–30,000 cells per well in 24-well plates onto Matrigel (354234; Corning) as previously described (*Lee et al., 2007*) and imaged using a Zeiss Axio Vert.A1 inverted confocal microscope after nine days. Alternatively, cells were plated at 20,000 cells per well in 24-well on ultra low-attachment plates (Corning) in DMEM F12 supplemented with 1X B27 (17504044; Life Technologies), 20 ng/mL EGF (GF144; EMD Millipore) and 20 ng/mL bFGF (13256029; Life Technologies), and imaged after 5–18 days with 200 µL fresh media added every 5 days.

### Constructs

cDNA encoding C-terminal 3xFlag tagged human GLI1, GLI2, GLI3 and GLI2 ΔN were purchased from Addgene (#84922, 84920, 84921, 17649, respectively) and cloned into pMSCV retroviral vector or pRetroX-Tight-Pur doxycycline-inducible vector (generously provided by Dr. Nabeel Bardeesy; Massachusetts General Hospital). Mouse Gli2 was generously provided by Dr. Ryan Corcoran (Massachusetts General Hospital). Gli-luciferase and *Renilla* reporters were gifts from Dr. Jeremy Reiter (UCSF).

### Lentiviral-mediated knockdown

All shRNA were obtained from Sigma Aldrich in the pLKO vector and sequences are listed in *Supplementary file 1*. HEK293T cells were transfected with lentiviral or retroviral plasmids and packaging constructs using X-tremeGENE transfection (6365787001; Sigma Aldrich) reagent according to the manufacturer's instructions and as previously described (*Perera et al., 2015*). Cells were infected with virus-containing media using Polybrene reagent (TR-1003-G; EMD Millipore) according to the manufacturer's instructions, and selected for 48 hr in 2 µg/mL of puromycin or blasticidin.

## Luciferase reporter assay

Cells were plated at 100,000 cells per well in 12-well or 24-well plates and co-transfected with Gli-luciferase and *Renilla* luciferase plasmids using Lipofectamine 2000 (11668019; Life Technologies) according to the manufacturer's instructions. Cells were lysed and assayed for activity using the Dual-Luciferase Assay kit (E1960; Promega) 48 hr post-transfection.

## ELISA and exogenous ligand treatments

Secreted OPN in the supernatant of cultured cells was measured using the Human Osteopontin Quantikine ELISA kit (DOST00; R and D systems) according to manufacturers protocol. For experiments where exogenous ligand was added, recombinant human OPN (1433-OP-050/CF; R and D Systems) was added at 1 μg/mL and cells assayed 2–3 days later.

## Drug treatment

Cells were treated with 5 μM of GANT61 (3191/10; R and D Systems) for 3–5 days prior to collection of RNA or protein lysates. Alternatively, iKRAS cells were plated at 150,000 cells with or without Dox (1 μg/mL) in the presence of absence of 5 μM GANT61 and RNA was collected 72 hr post-plating.

## Immunoblotting

Cells were lysed in ice-cold lysis buffer (150 mM NaCl, 20 mM Tris [pH 7.5], 1 mM EDTA, 1 mM EGTA, 1% Triton X-100, 2.5 mM sodium pryophosphate, 1 mM β-glycerophosphate, 1 mM sodium vanadate, and one tablet of Pierce Protease Inhibitor Tablets, EDTA Free [Fisher Scientific-A32965] per 10 mL). Samples were clarified by sonication and centrifugation. Protein content was measured using Pierce BCA Protein Assay Kit (Life Technologies-23227), and 15–50 μg protein was resolved on 8–15% protein gels using SDS-PAGE and transferred onto PVDF membranes (EMD MIllipore-IPVH00010). Membranes were blocked in 5% non-fat dry milk (VWR-89406056) made up in Tris-buffered saline with 0.2% Tween 20 (TBS-T) prior to incubation with primary antibody overnight at 4°C in either 5% non-fat dry milk or 5% bovine serum albumin (BSA, Sigma Aldrich-A4503). Membranes were washed in TBS-T and developed after 45-min incubation in species-specific horseradish peroxidase-conjugated secondary antibody, and visualized using supersignal west pico chemiluminescent substrate (Fisher Scientific-34080), and imaged using the ChemiDoc XRS + System (Biorad).

## Immunofluorescence

Cells were plated on fibronectin-coated glass cover- slips at 100,000–300,000 cells per coverslip. Twelve-to-sixteen hours later, the slides were rinsed with PBS once and fixed for 15 min with 4% paraformaldehyde at room temperature prior to permeabilization with 0.1% Saponin for 5 min. Slides were incubated with primary antibody in 5% normal goat serum overnight at 4 degC, rinsed four times with PBS, and incubated with secondary antibodies produced in goat (diluted 1:400 in 5% normal goat serum) for 45 min at room temperature in the dark. Slides were mounted on glass slides using Vectashield (Vector Laboratories) and imaged on a Zeiss Laser Scanning Microscope (LSM) 710. Images were processed using ImageJ.

## RNA isolation and quantitative RT-PCR

Total cellular RNA was extracted using the PureLink RNA Mini Kit (12183025; Thermo Fisher). Reverse transcription was performed using the iScript Reverse Transcription Supermix (1708841; Bio-Rad) followed by quantitative RT–PCR with iTaq Universal SYBR Green Supermix (1725122; Bio-Rad) using the CFX384 Touch Real Time PCR Detection System (BioRad). Results are presented as the mean of at least three technical replicates and are representative of at least n = 3 biological replicates. Primer sequences are listed in *Supplementary file 1*.

## Generation of GLI2 and SPP1 knockout cell lines using CRISPR/Cas9

GLI2 and SPP1 knockouts in KP4 cells were generated using the RNP-electroporation method as previously described (*Liang et al., 2015*). One million KP4 cells were used per electroporation using the Amaxa 4D Nucleofector kit (V4XC-9064, Lonza). Guide RNA and Cas9 complexes were formed using 160 μM crRNA annealed to 160 μM tracrRNA (Dharmacon) and incubated with 40 μM Cas9 protein (purchased from University of California, Berkeley). Cutting efficiency was assessed 48 hr

post-electroporation using PCR and sanger sequencing. GLI2 and SPP1 knockout was confirmed using quantitative RT-PCR, immunoblotting and ELISA after clonal expansion of single cells.

Guide RNA sequences 5′–3′:
GLI2 exon 2 – TTTGGCTTCTTGCTTCTCGG
SPP1 exon 2 – GTATGGCACAGGTGATGCCT
PCR primer sequences 5′–3′:
GLI2 exon 2 FW – GTGAAGGAGTGAGCGAACATGC
GLI2 exon 2 RV – TCTTCGCCCTCCATAAACCCAG
SPP1 exon 2 FW – GCAAAATTTCCCTTTCCCTTGCC
SPP1 exon 2 RV – ACTGTGCTTGGTACTGGCCTAG

## Chromatin immunoprecipitation

293 T cells were transiently transfected with GLI2-Flag and fixed with 1% formaldehyde for 15 min and quenched with 125 mM glycine for 5 min. Cells were washed with cold PBS and collected as $10^7$ cell pellets. Cells were resuspended in 500 µL cold L1 buffer (50 mM Tris pH 8.0, 2 mM EDTA, 0.1% NP-40, 10% glycerol, 1 mM PMSF, 1x Pierce Protease Inhibitor) on ice for 5 min, centrifuged and resuspended in 450 µL cold SDS buffer (50 mM Tris pH 8.0, 10 mM EDTA, 1% SDS, 1 mM PMSF, 1x Pierce Protease Inhibitor). Chromatin was sheared for 10 cycles using the Bioruptor Pico to obtain fragments of 200–500 base pairs, and diluted 1:10 in cold ChIP buffer (0.5% NP-40, 5 mM EDTA, 200 mM NaCl, 50 µM Tris pH 8.0, 1 mM PMSF). Diluted chromatin was pre-cleared with 100 µL washed Protein A Dynabeads (Invitrogen) at 4°C for 1 hr then immunoprecipitated with Flag antibody (CST, 14793) or negative control Rabbit IgG (CST, 2729) at 4°C overnight. Immunocomplexes were recovered using 50 µL washed Protein A Dynabeads at 4°C for 2 hr. Beads were washed two times in following buffers: 800 µL Wash buffer (0.1% SDS, 1% NP-40, 2 mM EDTA, 500 mM NaCl, 20 mM Tris pH 8.0, 1 mM PMSF), LiCl buffer (0.1% SDS, 1% NP-40, 2 mM EDTA, 0.5M LiCl, 20 mM Tris pH 8.0, 1 mM PMSF) then TE buffer (1 mM EDTA, 10 mM Tris pH 8.0) for 5 min each all on ice. Complexes were eluted by resuspending beads in 100 µL 2% SDS in TE buffer then de-crosslinked overnight with 5 µL NaCl at 65°C. Recovered DNA was PCR purified and analyzed using qPCR. Primers were used to amplify a region that contains a GLI binding site in the *SPP1* promoter (−2324 to −2316) as reported in *Kijewska et al. (2017)*. Primers targeting the *ACTB* promoter served as a negative control while primers targeting a known GLI-binding site in the *BCL2* promoter (−957 to −949) served as a positive control (*Regl et al., 2004*).

ChIP qPCR primer sequences 5′–3′:
ACTB promoter FW – ATGCAGCGATCAGTGGCGT
ACTB promoter RV – TCCAGCTTCTTGTCACCACCTC
BCL2 promoter FW – CCGGACGCGCCCTCCC
BCL2 promoter RV – GGTGCCTGTCCTCTTACTTCATTCTC
SPP1 promoter FW – CTGACAGAAAATCCTACTCAGAAAA
SPP1 promoter RV – AAAGTAGGAAATGGATGCTGCG

## Subcutaneous and tail vein injections

For xenograft experiments, 4 million KP4 Cas9 control, GLI2$^{KO}$, or SPP1$^{KO}$ cells were injected into the flank of *NOD.SCID-Il2rg$^{-/-}$* (NSG) immunodeficient mice, and tumor volumes were measured using a caliper and calculated as tumor volume = ½ (length x width$^2$). For tail vein injections, 2 million cells were injected into the tail vein of NSG mice. End point criteria included poor body condition and weight loss. Sample sizes for in vivo experiments were calculated using a online tool (http://www.bu.edu/orccommittees/iacuc/policies-and-guidelines/sample-size-calculations/), taking into account variability of the assays and inter-individual differences within groups. In summary, we will use 8–10 mice per cohort (male and female) which provides statistical power of >0.8 ($\alpha$ = 0.05) to detect differences of approximately 50%, assuming a normal distribution.

## Histology and immunostaining

Tissue samples were fixed overnight in 10% formalin, and then embedded in paraffin and sectioned (5 mm thickness) by the UCSF mouse histopathology core. Haematoxylin and eosin staining was performed using standard methods. Slides were baked at 60°C for an hour, deparaffinized in xylenes

(three treatments, 5 min each), rehydrated sequentially in ethanol (5 min in 100%, 5 min in 90%, 5 min in 70%, 5 min in 50%, and 5 min in 30%), and washed for 5 min in water twice. For antigen unmasking, specimens were cooked in a 10 mM sodium citrate buffer (pH 6.0) for 10 min at 95°C using conventional pressure cooker, rinsed three times with PBS, incubated for 1 hr with 1% $H_2O_2$ at room temperature to block endogenous peroxidase activity, washed three times with PBS, and blocked with 2.5% goat serum in PBS for 1 hr. Primary antibodies were diluted in blocking solution as follows: anti-S100A2 (abcam, ab109494) 1:200; anti-GATA6 (CST, 5851) 1:500 and incubated with the tissue sections at 4°C overnight. Specimens were then washed three times for 5 min each in PBS and incubated with secondary anti-mouse/rabbit IgG (Vector Laboratories, MP-7500) for 1 hr at room temperature. Following three washes in PBS, slides were stained for peroxidase for 3 min with the DAB(di-aminobenzidine) substrate kit (SK-4100, Vector Laboratories), washed with water and counterstained with haematoxylin. Stained slides were photographed with a KEYENCE BZ-X710 microscope.

## RNAseq and GSEA

RNA sequencing data containing gene expression values from pancreatic ductal adenocarcinoma samples sequenced as part of the TCGA project was downloaded from the cBioPortal (http://www. cbioportal.org/index.do). The data are represented as z scores where zero represents the mean value of normalized and log transformed gene expression for samples diploid at that locus in the sample set, and a z score of one corresponding to expression one standard deviation above the mean. A subset of tumors corresponding to those with levels of *GLI2* expression that were high (z scrore >0.5; n = 51) or low (z score <−0.5; n = 41) were then analyzed for their corresponding gene expression values of the ligands *IHH* or *SHH* (*Figure 1—figure supplement 1E*). Assignment of PDA TCGA samples as classical or basal-like and extent of tumor cellularity was based on an independently published study (*Cancer Genome Atlas Research Network, 2017*). Overall survival (OS) was defined as the time of surgery to the date of death from any cause. Disease-free survival (DFS) was defined as the date of surgery to the date of tumor recurrence at any site or to the date of last follow-up. Parameters for determining OS and DFS were defined as z score > −1 for *GLI2* high status, z score >2 for *SHH* high status (*Figure 1D,E*) and z score >0.5 for *SPP1* high status (*Figure 7K*). OS and DFS were analyzed using Kaplan–Meier and log-rank tests. Significance was determined as a *P* value < 0.05.

Data for the YAPC-iEV and YAPC-iGLI2 cells in *Figure 3D* was processed using a standard RNA-seq pipeline that used Trimmomatic to clip and trim the reads, tophat2 to align the reads to hg19, and cuffdiff to calculate differential expression. GSEA (http://www.broadinstitute.org/gsea/index.jsp) of the expression data was used to assess enrichment of the epithelial-to-mesenchymal gene signature. The Moffitt (*Moffitt et al., 2015*) basal-like gene signature was used to assign basal-like status. For the classical signature, a combination of genes from the Collisson classical (*Collisson et al., 2011*), Moffitt classical (*Moffitt et al., 2015*) and Bailey progenitor (*Bailey et al., 2016*) signatures were used (See *Supplementary file 2*).

## Image and statistical analysis

Image analysis, including densitometry and spheroid quantification was conducted using Image J software (NIH). Statistical analyses of results are expressed as mean ± standard deviation unless otherwise indicated. For each box-and-whisker plot, center line is the median and whiskers represent the minimum and maximum values. Significance was analyzed using two-tailed Student's t-test and Log-rank (Mantel-Cox) test for survival data. A p value of less than 0.05 was considered statistically significant. Graphing and statistical analyses were performed with GraphPad Prism seven software. All experiments were performed at least three times or in at least two separate cell lines. Data is displayed as either representative or the average of 2–4 independent biological replicates with at least three technical replicates per experiment.

Correlations between demographic and clinical factor variables and the genes of interest, GLI2 and SHH were performed. Correlations were stratified based on expression levels (previously defined with z score thresholds). Significance was analyzed using a Chi squared test for categorical variables (sex, TNM staging), Fisher's exact test for dichotomous categorical variables (GLI2 and

SHH expression), and a two sample t-test for continuous variables (age and mutation count). A p-value<0.05 was considered a statistically significant association.

Cox proportional hazards survival analysis regression models were used to identify demographic and clinical factors with significant predictive value for the outcomes of overall survival and progression free survival. Factors included in the analysis were age at diagnosis, sex, TNM staging, mutation count, and GLI2 and SHH expression levels previously defined based on z-scores. Unstratified models and models stratified based on GLI2 and SHH expression levels were performed. A p-value<0.05 was considered a statistically significant association. Differences in sample sizes (n) attributed to missing clinical data. These analyses and figures were generated using R statistical software.

## Acknowledgements

We thank E Collisson, L Selleri, D Raleigh and R Zoncu for advice and helpful comments on the manuscript and F Kottakis and J Yano for bioinformatics and technical support, respectively. RMP is the Nadia's Gift Foundation Innovator of the Damon Runyon Cancer Research Foundation (DRR-46–17). This work is additionally supported by an NIH Director's New Innovator Award (1DP2CA216364) and Pancreatic Cancer Action Network Career Development Award to RMP and a National Science Foundation Graduate Research Fellowship and an NIH F31 Ruth L Kirschstein National Research Service Award to CRA.

## Additional information

### Funding

| Funder | Grant reference number | Author |
|---|---|---|
| National Cancer Institute | Direcetor's New Innovator Award (1DP2CA216364) | Rushika M Perera |
| Pancreatic Cancer Action Network | Career Development Award (16-20-25-PERE) | Rushika M Perera |
| Damon Runyon Cancer Research Foundation | Nadia's Gift Foudnation Innovator of the Damon Runyon Cancer Research Foundation (DRR-46-17) | Rushika M Perera |
| Hirshberg Foundation for Pancreatic Cancer Research | | Rushika M Perera |
| National Science Foundation | Graduate Research Fellowship | Christina R Adams |
| National Cancer Institute | Ruth L. Kirschstein National Research Service Award (1F31CA224792) | Christina R Adams |

The funders had no role in study design, data collection and interpretation, or the decision to submit the work for publication.

### Author contributions

Christina R Adams, Conceptualization, Data curation, Investigation, Methodology, Writing—original draft; Htet Htwe Htwe, Timothy Marsh, Data curation, Investigation, Methodology; Aprilgate L Wang, Megan L Montoya, Data curation, Investigation; Lakshmipriya Subbaraj, Aaron D Tward, Formal analysis, Statistical analysis of patient PDA cohorts; Nabeel Bardeesy, Resources, Writing—review and editing; Rushika M Perera, Conceptualization, Data curation, Formal analysis, Supervision, Funding acquisition, Investigation, Methodology, Writing—original draft, Writing—review and editing

## Author ORCIDs

Christina R Adams (iD) https://orcid.org/0000-0003-3490-6444
Timothy Marsh (iD) https://orcid.org/0000-0002-8772-7645
Rushika M Perera (iD) https://orcid.org/0000-0003-2435-2273

## Ethics

Animal experimentation: Studies were performed in accordance with approved Institutional Animal Care and Use Committee (IACUC) protocols (#AN132973), which are reviewed annually. The UCSF animal management program is accredited by the American Association for the Accreditation of Laboratory Animal Care (AAALAC), and meets National Institutes of Health standards as set forth in the "Guide for the Care and Use of Laboratory Animals." The Office of Laboratory Animal Welfare Assurance Number is A3400-01.

## Decision letter and Author response

Decision letter https://doi.org/10.7554/eLife.45313.029
Author response https://doi.org/10.7554/eLife.45313.030

# Additional files

## Supplementary files

• Supplementary file 1. Human and mouse primer sequences used in the study.
DOI: https://doi.org/10.7554/eLife.45313.023

• Supplementary file 2. Basal-like and classical gene signatures. List of genes associated with the classical and basal-like gene signatures and their expression in the corresponding Moffitt, Collisson and Bailey signatures.
DOI: https://doi.org/10.7554/eLife.45313.024

• Transparent reporting form
DOI: https://doi.org/10.7554/eLife.45313.025

## Data availability

Sequencing data from Figure 3 have been deposited in GEO under accession code GSE131222.

The following dataset was generated:

| Author(s) | Year | Dataset title | Dataset URL | Database and Identifier |
|---|---|---|---|---|
| Adams CR, Htwe HH, Marsh T, Wang AL, Montoya ML, Tward AD, Bardeesy N, Perera R | 2019 | Gene expression changes associated with induction of GLI2 in human PDA cells | http://www.ncbi.nlm.nih.gov/geo/query/acc.cgi?acc=GSE131222 | NCBI Gene Expression Omnibus, GSE131222 |

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
