## [Decision Letter]

Thank you for submitting your article "Transcriptional control of subtype switching ensures adaptation and growth of pancreatic cancer" for consideration by *eLife*. Your article has been reviewed by two peer reviewers, including Wilbert Zwart as the Reviewing Editor and Reviewer #1, and the evaluation has been overseen by Jeffrey Settleman as the Senior Editor. The following individual involved in review of your submission has also agreed to reveal his identity: Claus Jørgensen (Reviewer #2).

The reviewers have discussed the reviews with one another and the Reviewing Editor has drafted this decision to help you prepare a revised submission.

Summary:

Here, Perera and colleagues describe a novel role of *GLI2* in regulating subtype switching, EMT status and RAS dependency in pancreatic cancer.

Pancreatic cancer cells can exist as different subtypes, and this has previously been linked to patient performance and therapeutic sensitivity. However, while there has been a lot of focus on defining subtypes and their potential role in PDA, less is known about the regulators of PDA subtypes. The data presented describing a role for *GLI2* in regulating PDA subtype switching are rigorous, experiments are well conducted, key controls are in place and the manuscript is logical and well presented.

Major points:

1) Figure 1D,E: multivariate correction with clinical parameters should be performed; is the difference in survival related to the gene of interest, or to other clinical variables that associate with these factors? Also, do the 2 arms between the different KM curves identify the same patient populations? Are the patients with *GLI2* high the same as those with SSH low?

2) Figure 3: the text states that two cell lines were used in these analyses (HPAFII and YAPC), for I can only find the data (especially 3B and D) for the YAPC cell lines. Please confirm findings in several cell lines, and include those data in the paper.

3) Figure 4: same thing here as in Figure 3. In the text, two cell lines are mentioned (KP4 and Panc0327), but the data presented don't show the findings in both cell lines, only one. Please include the data for both cell lines.

4) In Figure 3B, 3D growth analyses (not organoids) is depicted with *GLI2* over expression, observing effects on migration. Wouldn't you also expect to see impact on migratory phenotype in Figures 5H and K, when overexpressing *GLI2* (H) or the opposite when knocking it down (K)?

---

## [Author Response]

Major points:1) Figure 1D,E: multivariate correction with clinical parameters should be performed; is the difference in survival related to the gene of interest, or to other clinical variables that associate with these factors? Also, do the 2 arms between the different KM curves identify the same patient populations? Are the patients with GLI2 high the same as those with SSH low?

We have performed univariate and multivariate analysis on the data presented in Figure 1D and E as requested. In a univariate analysis, neither *GLI2* high versus low expression nor SHH high versus low expression was correlated with clinical variables such as sex, T stage, N stage, or M stage, although, as expected from our prior data, there was a correlation between high SHH and low *GLI2* and vice-versa. These data have been included as new panels in Figure 1—figure supplement 1G,H of the revised manuscript. A multivariate analysis of the hazard attributable to SHH or *GLI2* status demonstrated that no other variables explained the relationship between SHH or *GLI2* and overall or progression free survival. Thus, we conclude that *GLI2*-high status is an independent prognostic indicator of poor outcomes in PDA. These data have been included as a new supplementary figure and are discussed in the updated text (new Figure 1—figure supplement 2; subsection “Expression of Hh ligands and GLI transcription factors are anti-correlated and predict survival outcomes in PDA”).

2) Figure 3: the text states that two cell lines were used in these analyses (HPAFII and YAPC), for I can only find the data (especially 3B and D) for the YAPC cell lines. Please confirm findings in several cell lines, and include those data in the paper.

We apologize for the oversight. We have now included data demonstrating *GLI2*-mediated basal-like switching in HPAFII cells (Figure 3—figure supplement 1A, C) and as requested by the reviewer, we provide new data in mouse PDA cells reinforcing these findings (Figure 5—figure supplement 2B). For clarity, we have indicated which cell lines have been used for each experiment within the manuscript text. Consistent with our original data in YAPC cells, ectopic expression of *GLI2* in HPAFII cells (HPAFII-*GLI2*) led to a change in cell morphology involving loss of a compact, cobblestone growth pattern and increased spreading (Figure 3—figure supplement 1C). Additionally, HPAFII-*GLI2* cells showed downregulation of epithelial markers, ESRP1, the transcription factor GATA6 – a regulator of the classical subtype of PDA, and SHH and induction of ZEB1 (Figure 3—figure supplement 1A). HPAFII-iGLI2 cells also showed upregulation of the basal-like gene signature and concomitant downregulation of the classical signature (Figure 3E), which are all consistent with data obtained in YAPC cells. Thus, we provide evidence in both YAPC and HPAFII cell lines indicating that ectopic *GLI2* expression is sufficient to drive a switch from a classical to a basal-like state.

3) Figure 4: same thing here as in Figure 3. In the text, two cell lines are mentioned (KP4 and Panc0327), but the data presented don't show the findings in both cell lines, only one. Please include the data for both cell lines.

As requested by the Reviewer we provide data in both KP4 and Panc0327 cells to support our finding that *GLI2* is required to maintain a basal-like state in PDA. We have also indicated which cell lines have been used for each experiment within the manuscript text. CRISPRCas9 mediated knockout of *GLI2* in Panc0327 and KP4 cells caused a switch toward a more epithelial-like morphology in 2D monolayer culture (Figure 4D) and 3D Matrigel growth (KP4; Figure 4—figure supplement 1D). Additionally, *GLI2* KO in Panc0327 cells resulted in reduced expression of EMT markers ZEB1, VIM and CDH2 and the stemness factor *SOX2* (Figure 4—figure supplement 1C), a decrease in basal-like markers KRT14 and KRT5, and an increase in classical markers ESRP1 and GATA6 (Figure 4—figure supplement 1A) and a coordinated reduction in expression of the basal-like gene signature (Figure 4—figure supplement 1B), consistent with results obtained with KP4 cells (Figure 4A-C). Finally, immunofluorescence staining of Panc0327 and KP4 cells following knockout or knockdown of *GLI2* respectively, caused a prominent nuclear re-localization of ESRP1 (Figure 4E and Figure 4—figure supplement 1E). Together these data demonstrate a requirement for *GLI2* in maintaining a basal-like state in both Panc0327 and KP4 PDA cell lines.

4) In Figure 3B, 3D growth analyses (not organoids) is depicted with GLI2 over expression, observing effects on migration. Wouldn't you also expect to see impact on migratory phenotype in Figures 5H and K, when overexpressing GLI2 (H) or the opposite when knocking it down (K)?

We thank the reviewers for pointing this out. We have found close concordance between the human and murine cell systems regarding the *GLI2*-mediated subtype switching, with the sole exception being the lack of changes in cell migration in the mouse cells as noted by the reviewer. For example, consistent with the data in human cells, ectopic *GLI2* expression in mouse iKRAS4 cells (iKRASGLI2) induced expression of mesenchymal markers ZEB1 and N-Cadherin, basal markers KRT14 and KRT5, and *SOX2* (Figure 5—figure supplement 2B), and upregulated the basallike gene signature (compare Figure 5—figure supplement 1D and Figure 5E). We provide new data showing that expression of *GLI2* in YAPC-*GLI2* cells is able to rescue growth following suppression of oncogenic KRAS (Figure 5—figure supplement 2D) similar to that observed in iKRASGLI2 mouse PDA cells (Figure 5—figure supplement 2D). As previously observed, YAPC-*GLI2* cells additionally show a striking morphological change when grown in 3D culture that is retained following KRAS loss (Author response image 1 bottom panel: images show the effect of stable expression of EV (top) or GLI2 (bottom) on YAPC sphere formation in response to shRNAmediated KRAS knockdown).

These data support a conserved function for *GLI2* in driving a basal-like subtype switch and resistance to mutant KRAS loss in human and mouse PDA cells.